# Does Graph Distillation See Like Vision Dataset Counterpart?

**Beining Yang**[1,2]\*, **Kai Wang**[3]\*, **Qingyun Sun**[1,2]†, **Cheng Ji**[1,2], **Xingcheng Fu**[1,2], **Hao Tang**[4], **Yang You**[3], **Jianxin Li**[1,2]‡

[1]School of Computer Science and Engineering, Beihang University
[2]Advanced Innovation Center for Big Data and Brain Computing, Beihang University
[3]National University of Singapore    [4]Carnegie Mellon University

## Abstract

Training on large-scale graphs has achieved remarkable results in graph representation learning, but its cost and storage have attracted increasing concerns. Existing graph condensation methods primarily focus on optimizing the feature matrices of condensed graphs while overlooking the impact of the structure information from the original graphs. To investigate the impact of the structure information, we conduct analysis from the spectral domain and empirically identify substantial Laplacian Energy Distribution (LED) shifts in previous works. Such shifts lead to poor performance in cross-architecture generalization and specific tasks, including anomaly detection and link prediction. In this paper, we propose a novel **S**tructure-broadcasting **G**raph **D**ataset **D**istillation (**SGDD**) scheme for broadcasting the original structure information to the generation of the synthetic one, which explicitly prevents overlooking the original structure information. Theoretically, the synthetic graphs by SGDD are expected to have smaller LED shifts than previous works, leading to superior performance in both cross-architecture settings and specific tasks. We validate the proposed SGDD across 9 datasets and achieve state-of-the-art results on all of them: for example, on the YelpChi dataset, our approach maintains 98.6% test accuracy of training on the original graph dataset with 1,000 times saving on the scale of the graph. Moreover, we empirically evaluate there exist $17.6\% \sim 31.4\%$ reductions in LED shift crossing 9 datasets. Extensive experiments and analysis verify the effectiveness and necessity of the proposed designs. The code is available in the https://github.com/RingBDStack/SGDD.

## 1   Introduction

Graphs have been applied in many research areas and achieved remarkable results, including social networks [27, 67, 73, 79], physical [16, 3, 57, 10], and chemical interactions [2, 93, 111, 80]. Graph neural networks (GNNs), a classical and wide-studied graph representation learning method [40, 84, 98, 63], is proposed to extract information via modeling the features and structures from the given graph. Nevertheless, the computational and memory costs are extremely heavy when training on a large graph [48, 89, 99]. One of the most straightforward ideas is to reduce the redundancy of the large graph. For example, graph sparsification [66, 77] and coarsening [53, 52, 46, 75] are proposed to achieve this goal by dropping redundant edges and grouping similar nodes. These methods have shown promising results in reducing the size and complexity of large graphs while preserving their essential properties.

---

\*Equal contribution (yangbeining@buaa.edu.cn, kai.wang@comp.nus.edu.sg).
†Project lead sunqy@buaa.edu.cn
‡Corresponding author lijx@buaa.edu.cn.

37th Conference on Neural Information Processing Systems (NeurIPS 2023).

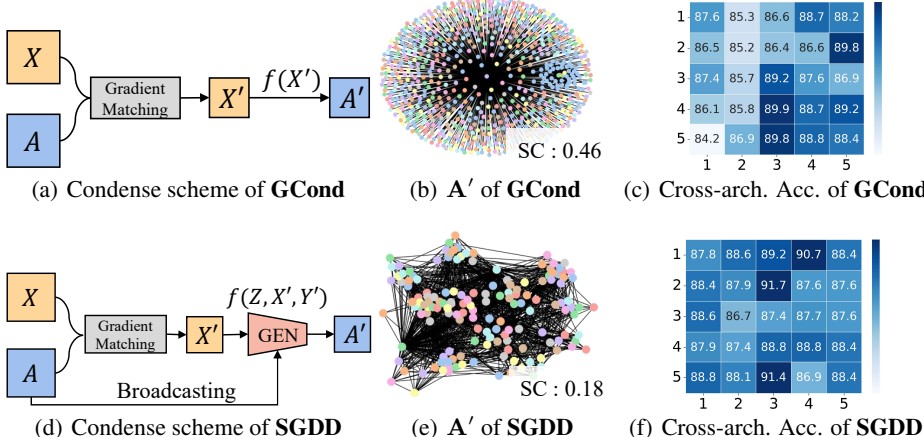

Figure 1: (a) and (d) illustrate that the pipelines of SGDD and GCond [37]. One can find that SGDD broadcast $\mathbf{A}$ from the original graph to the generation of $\mathbf{A}'$ while GCond synthesizes $\mathbf{A}'$ via the operation $f(\cdot)$ (*e.g.*, pair-wise feature similarity [74]) on $\mathbf{X}'$. We also show the condensed graph and its shift coefficient (SC) of GCond and SGDD in (b) and (e), respectively. Note that we introduce $SC$ as an approximation of the LED shift. We show the cross-architecture performance of GCond and SGDD in (c) and (f), 1, 2, 3, 4, and 5 denote APPNP [42], Cheby [13], GCN [41], SAGE [26], and SGC [92], more cross-architecture results can be found in Tab. 2.

However, graph sparsification and coarsening heavily rely on heuristics [5] (*e.g.*, the largest principle eigenvalues [78], pairwise distances [66, 81]), which may lead to poor generalization on various architectures or tasks and sub-optimal for the downstream GNNs training [37]. Most recently, as shown in Fig. 1(a), GCond [37] follows the vision dataset distillation methods [88, 105, 24, 103] and proposes to condense the original graph by the gradient matching strategy. The feature of the synthetic graph dataset is optimized by minimizing the gradient differences between training on original and synthetic graph datasets. Then, the synthetic feature is fed into the function $f(\mathbf{X}')$ (*e.g.*, pair-wise feature similarity [74]) to obtain the structure of the graph. The synthetic graph dataset (including feature and structure) is expected to preserve the task-relevant information and achieve comparable results as training on the original graph dataset at an extremely small cost.

Although the gradient matching strategy has achieved significant success in graph dataset condensation, most of these works [37, 36, 50, 109] follow the previous vision dataset distillation methods to synthesize the condensed graphs, which results in the following limitations: 1) In Fig. 1(b), we can find that the original graph structure information is not well preserved in the condensed graph. That's because they build the structure ($\mathbf{A}'$) upon the learned feature ($\mathbf{X}'$) (Fig. 1(a)), which may cause the loss of original structure information. 2) In vision dataset distillation, applying the gradient matching strategy may result in an entanglement between the synthetic dataset and the architecture that is used for condensing [106, 104, 39]. This can detrimentally impact their performance, especially when it comes to generalizing to unseen architectures [22]. This issue is compounded when dealing with graph data that comprise both features and structures. As shown in Fig. 1(c) and Tab. 2, GCond [37] explicitly displays inferior performance, demonstrating that generating structure only based on the feature degrades the generalization of the condensed graph.

To investigate the relation between the structure of the condensed graph and the performance of training on it, we follow previous works [82, 1, 13, 78] to analyze it from a spectral view. Specifically, we explore the relationship between the Laplacian Energy Distribution (LED) shift [82, 25, 12] and the generalization performance of the condensed graph. We empirically find a positive correlation between LED shift and the performance in cross-architecture settings. To address these issues, we introduce a novel Structure-broadcasting Graph Dataset Distillation (SGDD) scheme to condense the original graph dataset by broadcasting the original adjacency matrix to the generation of the synthetic one, named SGDD, which explicitly prevents overlooking the original structure information. As shown in Fig. 1(d), we propose the graphon approximation to broadcast the original structure $\mathbf{A}$ as supervision to the generation of condensed graph structure $\mathbf{A}'$. Then, we optimize $\mathbf{A}'$ by minimizing

the optimal transport distance of these two structures. For $\mathbf{X}'$, we follow the [37] to synthesize the feature of the condensed graph. The $\mathbf{A}'$ and $\mathbf{X}'$ are jointly optimized in a bi-level loop. Both theoretical analysis (Sec. 3.2) and empirical studies (Fig. 1(c) and 1(f)) consistently show that our method reduces the LED shift significantly ($0.46 \rightarrow 0.18$).

We further evaluate the impact of LED shifts and conduct experiments in the cross-architecture setting. Comparing Fig. 1(f), 1(c), and Tab. 2, the improvement of SGDD is up to 11.3%. To evaluate the generalization of our method, we extend SGDD to node classification, anomaly detection, and link prediction tasks. SGDD achieves state-of-the-art results on these tasks in most cases. Notably, we obtain new state-of-the-art results on YelpChi and Amazon with 9.6% and 7.7% improvements, respectively. Our main contributions can be summarized as follows:

- Based on the analysis of the difference between graph and vision dataset distillation, we introduce SGDD, a novel framework for graph dataset distillation via broadcasting the original structure information to the generation of a condensed graph.

- In SGDD, the graphon approximation provides a novel scheme to broadcast the original structure information to condensed graph generation, and the optimal transport is proposed to minimize the LED shifts between original and condensed graph structures.

- SGDD effectively reduces the LED shift between the original and the condensed graphs, consistently surpassing the performance of current state-of-the-art methods across a variety of datasets.

## 2 Related Work

**Dataset Distillation & Dataset Condensation.** Dataset distillation (DD) [88, 106, 24, 51, 110, 23, 86, 49, 87, 6, 7] aims to distill a large dataset into a smaller but informative synthetic one. The proposed method imposes constraints on the synthetic samples by minimizing the training loss difference, while ensuring that the samples remain informative. This technique is useful for applications such as continual learning [105, 106, 39, 44, 104], as well as neural architecture search [61, 62, 96]. Recently, GCond [37] is proposed to reduce a large-scale graph to a smaller one for node classification using the gradient matching scheme in DC [105]. Unlike GCond [37], who build the structure only through the learned feature, we explicitly broadcast the original graph structure to the generations to prevent the overlooking of the original graph structure information.

**Graph Coarsening & Graph Sparsification.** Graph Coarsening [53, 52, 14] follows the intuition that nodes in the original graph can naturally group the similiar node to a super-nodes. Graph Sparsification [66, 38, 78] is aimed at reducing the edges in the original graph, as there are many redundant relations in the graphs. We can simply sum up both two methods by trying to reduce the "useless" component in the graph. Nevertheless, these methods primarily rely on unsupervised techniques such as the largest $k$ eigenvalue [66] or the multilevel incomplete LU factorization [14], which cannot guarantee the behavior of the synthetic graph in downstream tasks. Moreover, they fail to reduce the graph size to an extremely small degree (*e.g.*, reduce the number of nodes to 0.1% of the original). In contrast, graph condensation can aggregate information into a smaller yet informative graph in a supervised way, thereby overcoming the aforementioned limitations.

## 3 Preliminary and Analysis

### 3.1 Formulation of graph condensation

Consider a graph dataset $\mathcal{G} = \{\mathbf{A}, \mathbf{X}, \mathbf{Y}\}$, where $\mathbf{A} \in \mathbb{R}^{N \times N}$ denotes the adjacency matrix, $\mathbf{X} \in \mathbb{R}^{N \times d}$ is the feature, and $\mathbf{Y} \in \mathbb{R}^{N \times 1}$ represents the labels. $\mathcal{S} = \{\mathbf{A}', \mathbf{X}', \mathbf{Y}'\}$ is defined as the synthetic dataset, the first dimension of $\mathbf{A}'$, $\mathbf{X}'$, and $\mathbf{Y}'$ are $N'$ ($N' \ll N$).

The goal of graph condensation is achieving comparable results as training on the original graph dataset $\mathcal{G}$ via training on the synthetic one $\mathcal{S}$. The optimization process can be formulated as follow,

$$\mathcal{S}^* = \arg\min_{\mathcal{S}} \mathrm{M}(\theta_{\mathcal{S}}^*, \theta_{\mathcal{G}}^*) \quad \text{s.t} \quad \theta_t^* = \arg\min_{\theta} \mathrm{Loss}(\mathrm{GNN}_{\theta}(t)), \tag{1}$$

where $\mathrm{GNN}_{\theta}(t)$ denotes a GNN parameterized with $\theta$, $\theta_{\mathcal{S}}$ and $\theta_{\mathcal{G}}$ are the parameters that are trained on $\mathcal{S}$ and $\mathcal{G}$, and $t \in \{\mathcal{S}, \mathcal{G}\}$, $\mathrm{M}(\cdot)$ denotes a matching function and $\mathrm{Loss}(\cdot)$ is the loss function.

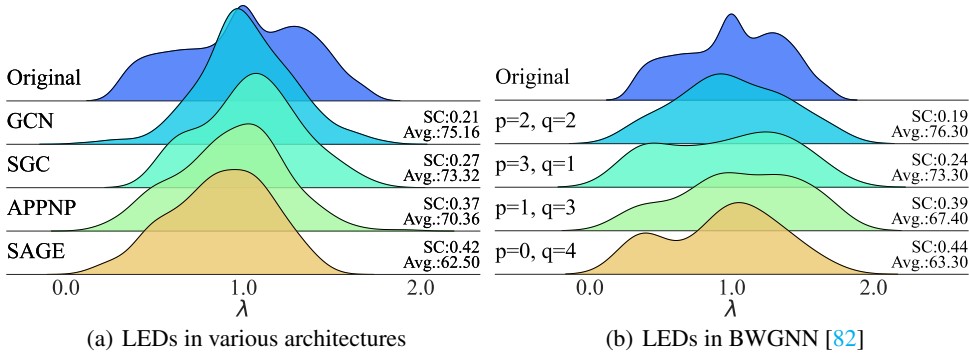

(a) LEDs in various architectures  (b) LEDs in BWGNN [82]

Figure 2: (a): Illustrations of the LEDs (use the density plotting, each peak represents the LED of a graph), SC, and Avg. (average test performances on GCN, SGC, APPNP, and SAGE) in various architectures. (b): Evaluation of these three metrics in frequency-adaptive BWGNN. We empirically find that the LED shifts (*i.e.*, SC) are high-consistent with Avg. performances, thus SC could be a good indicator to evaluate the performance of the condensed graph.

## 3.2 Analysis of the structure of the condensed graph and its effects

In this section, we first introduce the definition of Laplacian Energy Distribution (LED), then analyze the LED in previous work, and finally explore its influence on generalization performance.

**Laplacian Energy Distribution.** Following the previous works [25, 12, 82], we introduce Laplacian energy distribution (LED) to analyze the structure of graph data. Given an adjacency matrix $\mathbf{A} \in \mathbb{R}^{N \times N}$, the Laplacian matrix $\mathbf{L}$ is defined as $\mathbf{D} - \mathbf{A}$, where the $\mathbf{D}$ is the degree matrix of $\mathbf{A}$. We utilize the normalized Lapalacian matrix $\tilde{\mathbf{L}} = \mathbf{I} - \mathbf{D}^{-1/2}\mathbf{A}\mathbf{D}^{-1/2}$, where $\mathbf{I}$ is an identity matrix. The eigenvalues of $\tilde{\mathbf{L}}$ are then defined as $\lambda_1, \ldots \lambda_n, \ldots \lambda_N$ by ascending order, *i.e.*, $\lambda_1 \leq \cdots \lambda_n \leq \cdots \leq \lambda_N$, with orthonormal eigenvectors $U = (u_1, \cdots u_n, \cdots u_N)$ correspond to these eigenvalues.

**Definition 1** (Laplacian Energy Distribution [25, 12, 82]). *Let* $\mathbf{X} = (x_1, \cdots x_n, \cdots x_N)^\top \in \mathbb{R}^{N \times d}$ *be the feature of graph, we have* $\hat{\mathbf{X}} = (\hat{x}_1 \cdots \hat{x}_n \cdots \hat{x}_N)^\top = U^\top \mathbf{X}$ *as the post-Graph-Fourier-Transform of* $\mathbf{X}$. *The formulation of LED is defined as:*

$$\eta_n(\mathbf{X}, \tilde{\mathbf{L}}) = \frac{\hat{x}_n^2}{\sum_{i=1}^N \hat{x}_i^2} = \bar{x}_n. \tag{2}$$

**LED analysis of previous work.** First, we define $\eta^{\mathcal{G}}$ and $\eta^{\mathcal{S}}$ as the LED of the original graph and condensed graph, respectively. Then the LED shift of the above two graphs can be formulated as: $||\eta^{\mathcal{G}} - \eta^{\mathcal{S}}|| = ||\sum_{i=1}^N \eta_i(\mathbf{X}, \tilde{\mathbf{L}}) - \sum_{j=1}^{N'} \eta_j(\mathbf{X}', \tilde{\mathbf{L}}')||$. In GCond [37], the matching function M defaults as the MSE loss. Therefore, the objective of GCond can be written as $||\theta_{\mathcal{G}} - \theta_{\mathcal{S}}|| \leq \epsilon$, where $\epsilon$ is a small number as expected. Incorporating such an objective in the LED shift formula, the lower bound is shown as follows.

**Proposition 1.** *Refer to [1, 4, 82], the GNN can be recognized as the bandpass filter. Assume the frequency response area of GNN is* $(a, b)$, *where* $(a, b)$ *is architecture-specific. The lower bound of GCond is shown in Eq.* (3). *Detailed proof can be found in Appendix B.1.*

$$||\eta^{\mathcal{G}} - \eta^{\mathcal{S}}|| \geq \epsilon + \sum_{i=1}^a ||\bar{x}_i^2 - \bar{x}_i'^2|| + \sum_{j=b}^{N'} ||\bar{x}_j^2 - \bar{x}_j'^2||. \tag{3}$$

According to Eq. (3), we find the lower bond of LED in GCond is related to the frequency response area $(a, b)$ (*i.e.*, specific GNN). For example, GCN [40] or SGC [92] (utilized in GCond) is a low-pass filter [1], which emphasizes the lower Laplacian energy. As shown in Fig. 1(a), the $\mathbf{A}'$ is built upon the learned "low-frequency" feature $\mathbf{X}'$, which may fail to generalize to cross-architectures (*i.e.*, high-pass filters) and specific tasks.

**Exploration of the effects of LED shift on the generalization performance.** Although the LED shift phenomenon occurs in GCond, quantifying the LED shift is challenging because $\mathcal{G}$ and $\mathcal{S}$ have

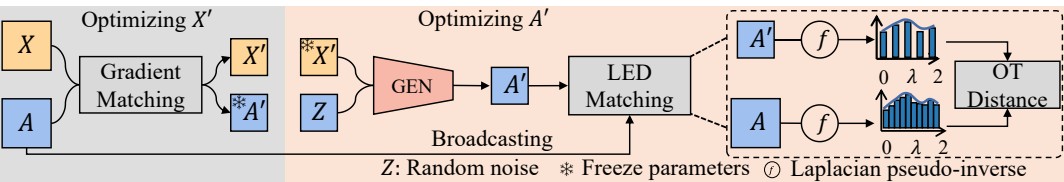

Figure 3: The Training pipeline of the SGDD (left). We first fix $\mathbf{A}'$ to optimize $\mathbf{X}'$ through the gradient matching strategy and we broadcast the supervision of $\mathbf{A}$ to the generation of the graph structure $\mathbf{A}'$. To mitigate the Laplacian Energy Distribution (LED) shift phenomenon, we propose the LED Matching strategy to optimize the $\mathbf{A}'$, which optimizes the learned structure with the optimal transport (OT) distance (right).

different numbers of nodes. To enable comparison, we follow an intuitive assumption that two nodes with similar eigenvalue distribution proportions can be aligned in comparison. Thus, we first convert LEDs of $\mathcal{G}$ and $\mathcal{S}$ into probability distributions using the Kernel Density Estimation (KDE) method [60]. Then we quantify the LED shifts as the distance of the probability distributions using the Jensen-Shannon (JS) divergence [47]. We define the LED shift coefficient ($SC$) in Definition 2:

**Definition 2** (LED shift coefficient, $SC$). *The LED shift coefficient between $\mathcal{G}$ and $\mathcal{S}$ is:*

$$SC = \text{JS}\left(\frac{1}{|V_\mathcal{G}|h}\sum_{\bar{x}_i \in X_\mathcal{G}} K(\frac{x-\bar{x}_i}{h})\Big|\Big|\frac{1}{|V_\mathcal{S}|h}\sum_{\bar{x}_j \in X_\mathcal{S}} K(\frac{x-\bar{x}_j}{h})\right), \qquad (4)$$

*where the $\text{JS}(\cdot||\cdot)$ denotes the JS-divergence, the $|V_\mathcal{G}|$ and the $|V_\mathcal{S}|$ is the number of the nodes to corresponding graphs, the $K(\cdot)$ represents the Gaussian kernel function with bandwidth parameter $h$. $SC \in [0,1]$ reflects the divergence between $\mathcal{G}$ and $\mathcal{S}$ (a smaller $SC$ indicates more similar).*

In Fig. 2, we empirically study the influences of $SC$ in two settings: various GNNs in Fig. 2(a) and fixed BWGNN [82] with adaptive bandpass in Fig. 2(b). **Based on the results in Fig. 2, We have several observations:** (1) The entangled learning paradigm that building structure (*i.e.*, adjacency matrix) upon on feature matrix will significantly lead to the LED shift phenomenon. (2) The positive correlation exists between the LED shift and the generalization performance of the condensed graph. (3) Preserving more information about the original graph structure may alleviate the LED shift phenomenon and improve the generalization performance of the condensed graph.

## 4 Structure-broadcasting Graph Dataset Distillation

In this section, we first present the pipeline and overview of SGDD in Fig. 3. Then, we introduce two modules of SGDD. Finally, we summarize the training pipeline of our SGDD.

### 4.1 Learning graph structure via graphon approximation

To prevent overlooking the original structure $\mathbf{A}$ from $\mathcal{G}$, we broadcast $\mathbf{A}$ as supervision for the generation of $\mathbf{A}'$. Considering the different shapes between $\mathbf{A}$ and $\mathbf{A}'$ ($N' \ll N$), we introduce graphon [21, 70, 17, 33, 94] to distill the original structure information to the condensed structure $\mathbf{A}'$. Specifically, given random noise $\mathcal{Z}(N') \in \mathbb{R}^{N' \times N'}$ as the input coordinates, through the generative model, we then synthesize an adjacency matrix $\mathbf{A}'$ with $N'$ nodes. This process can be formulated as $\mathbf{A}' = \text{GEN}(\mathcal{Z}(N'); \Phi)$, where the $\text{GEN}(\cdot)$ is a generative model with parameter $\Phi$, and the optimization process is then defined as:

$$\mathcal{L}_{\textbf{structure}} = \text{Distance}(\mathbf{A}, \text{GEN}(\mathcal{Z}(N'); \Phi)), \qquad (5)$$

where $\mathbf{A}$ is supervision and $\text{Distance}(\cdot)$ is a metric that measure the difference of $\mathbf{A}$ and $\mathbf{A}'$. The details of $\text{Distance}(\cdot)$ can be found in Sec. 4.2. To avoid overlooking of the inherent relation [32, 73] between $\mathbf{A}'$ and the corresponding node information (*i.e.*, $\mathbf{X}'$ and $\mathbf{Y}'$), we jointly input $\mathbf{X}'$ and $\mathbf{Y}'$ as conditions to generate $\mathbf{A}'$. Therefore, the final version of the generative model can be written as $\mathbf{A}' = \text{GEN}(\mathcal{Z}(N') \oplus \mathbf{X}' \oplus \mathbf{Y}'; \Phi)$, the $\oplus$ denotes the concatenate operation.

To study the performance of SGDD in the above paradigm, we theoretically prove the upper bound of LED shift in SGDD by invoking graphon theory. The result can be presented as follows.

**Proposition 2.** *The upper bound of the LED shift on SGDD is shown as:*

$$||\eta^{\mathcal{G}} - \eta^{\mathcal{S}}|| \leq \delta_{\square}(W_{\mathbf{A}}, \mathbf{A}'), \tag{6}$$

*where $\delta_{\square}$ denotes the cut distance [54] and $W_{\mathbf{A}}$ is the graphon of $\mathbf{A}$. See details in Appendix B.2.*

Note minimizing the upper bound of Eq. (6) is equal to optimizing the $L_{structure}$ on Eq. (5). Compared to the lower bound in (Eq. (3)), our upper bound is not related to any frequency response of specific GNN (*i.e.*, the terms of $\sum_{i=1}^{a} ||\bar{x}_i^2 - \bar{x}_i'^2|| + \sum_{i=b}^{N'} ||\bar{x}_i^2 - \bar{x}_i'^2||$). As a result, SGDD may perform better than the previous work, especially in the cross-architecture setting and specific tasks.

## 4.2 Optimizing the graph structure via optimal transport

To mitigate the LED shift between $\mathbf{A}$ and $\mathbf{A}'$, ideally, we can directly minimize the proposed $SC$. However, $SC$ requires an extremely time-consuming ($O(N^3)$) [4, 58] eigenvalue decomposition operation. Therefore, we propose the LED Matching strategy based on the *optimal transport* theory to form an efficient optimizing process.

Recall in the Eq.(4) of calculating $SC$, we first decompose the eigenvalue, followed by aligning the node through the JS divergence, which essentially compares the distribution proportions. The key point is to decide the node mapping strategy to align such two graphs. Alternatively, assuming we know the prior distribution of the alignment of $\mathcal{S}$ (*i.e.*, we know the bijection of nodes in $\mathcal{S}$ to the $\mathcal{G}$) and denoting such alignment as $\mathcal{S}*$, we can directly measure the distance between $\mathcal{G}$ and $\mathcal{S}*$ by employing the 2-Wasserstein metric[15, 56].

$$||\eta^{\mathcal{G}} - \eta^{\mathcal{S}*}|| = W_2^2\left(\eta^{\mathcal{G}}, \eta^{\mathcal{S}*}\right) \tag{7}$$

Furthermore, following the assumption in previous work[56, 15], we have $\eta^G \sim \mathcal{N}\left(0, L_{\mathcal{G}}^{\dagger}\right)$ and $\eta^S \sim \mathcal{N}\left(0, L_{\mathcal{S}*}^{\dagger}\right)$. Then, the Eq.(7) have a closed-form expression[15] as follows:

$$||\eta^{\mathcal{G}} - \eta^{\mathcal{S}*}|| = N\operatorname{tr}\left(L_{\mathcal{G}}^{\dagger}\right) + N'\operatorname{tr}\left(L_{\mathcal{S}*}^{\dagger}\right) - 2\operatorname{tr}\left(\sqrt{L_{\mathcal{S}*}^{\frac{\dagger}{2}} L_{\mathcal{G}}^{\dagger} L_{\mathcal{S}*}^{\frac{\dagger}{2}}}\right), \tag{8}$$

the $L^{\dagger}$ denotes the Laplacian pseudo-inverse operation and the $\operatorname{tr}$ indicates the trace operation of matrix. Therefore, even though we could not know the actual mapping strategy $\mathcal{S}*$, we can use the infimum of all possible strategies as a proxy solution. Formally, following the prior work [56], we employ the function $T$ as a transport plan in the metric space $\mathcal{X}$ to represent all feasible mapping strategies. Then, we use the $T_{\#}\eta^{\mathcal{S}}$ to represent the pushing forward process of transferring the distribution of $\eta^{\mathcal{S}}$ to the $\eta^{\mathcal{G}}$. As a result, the distance can be regarded as finding the infimum.

$$\begin{aligned}||\eta^{\mathcal{G}} - \eta^{\mathcal{S}}|| &= \inf_{T_{\#}\eta^{\mathcal{S}}=\eta^{\mathcal{G}}} \int_{\mathcal{X}} ||x - T(x)||^2 d\eta^{\mathcal{S}}(x) \\ &= N'\operatorname{tr}(L_{\mathcal{S}}^{\dagger}) - 2\operatorname{tr}\left(\left((L_{\mathcal{S}}^{\dagger})^{1/2} P^T L_{\mathcal{G}}^{\dagger} P \left(L_{\mathcal{S}}^{\dagger}\right)^{1/2}\right)^{1/2}\right).\end{aligned} \tag{9}$$

Here, due to the transport plan $T$ is impractical in optimizing, following the previous work[15], we use $P \in \mathbb{R}^{N' \times N}$ denotes as a free parameter serving as the direct mapping strategy between nodes. Thus, the $\mathrm{Distance}$ in the Eq. (5) could be directly optimized by the Eq.(9) (*i.e.*, use the $P$ represents all possible mapping strategies, the optimizing of $P$ is equal to choosing a more optimal mapping strategy). In the experimental setting, we use the Sinkhorn-Knopp[9] algorithm to optimize $P$.

The overall time complexity is reduced to the $O(N^{\omega}) \leq O(N^{2.373})$[15]. Note that the $L_{\mathcal{G}}^{\dagger}$ may be too large for computing, so we empirically sample a medium size (*e.g.*, 2,000 nodes) sub-structure in the experiment and ablate its influence in Appendix C.7.

Table 1: Comparisons to state-of-the-art methods. SGDD achieves the highest results in most cases on node classification (NC), anomaly detection (AD), and link prediction (LP) tasks. We report test accuracy (%) on NC datasets (including Citeseer, Cora, Ogbn-arxiv, Flickr, and Reddit), F1-macro (%) on AD datasets (including YelpChi and Amazon), and AUC (%) on LP datasets (including Citeseer-L and DBLP). **Bold entries** are best results, underline mark the runner-ups.

| | Dataset | Ratio ($r$) | Random | Herding | K-Center | Coarsening | GDC | Gcond | SGDD | Whole |
|---|---|---|---|---|---|---|---|---|---|---|
| NC | Citeseer [41] | 0.90% | $54.4_{\pm4.4}$ | $57.1_{\pm1.5}$ | $52.4_{\pm2.8}$ | $52.2_{\pm0.4}$ | $66.8_{\pm1.5}$ | $\mathbf{70.5}_{\pm1.2}$ | $\underline{69.5}_{\pm0.4}$ | $71.7_{\pm0.1}$ |
| | | 1.80% | $64.2_{\pm1.7}$ | $66.7_{\pm1.0}$ | $64.3_{\pm1.0}$ | $59.0_{\pm0.5}$ | $66.9_{\pm0.9}$ | $\mathbf{70.6}_{\pm0.9}$ | $\underline{70.2}_{\pm0.8}$ | |
| | | 3.60% | $69.1_{\pm0.1}$ | $69.0_{\pm0.1}$ | $69.1_{\pm0.1}$ | $65.3_{\pm0.5}$ | $66.3_{\pm1.5}$ | $\underline{69.8}_{\pm1.4}$ | $\mathbf{70.3}_{\pm1.7}$ | |
| | Cora [41] | 1.30% | $63.6_{\pm3.7}$ | $67.0_{\pm1.3}$ | $64.0_{\pm2.3}$ | $31.2_{\pm0.2}$ | $67.3_{\pm1.9}$ | $\underline{79.8}_{\pm1.3}$ | $\mathbf{80.1}_{\pm0.7}$ | $81.2_{\pm0.2}$ |
| | | 2.60% | $72.8_{\pm1.1}$ | $73.4_{\pm1.0}$ | $73.2_{\pm1.2}$ | $65.2_{\pm0.6}$ | $67.6_{\pm3.5}$ | $\underline{80.1}_{\pm0.6}$ | $\mathbf{80.6}_{\pm0.8}$ | |
| | | 5.20% | $76.8_{\pm0.1}$ | $76.8_{\pm0.1}$ | $76.7_{\pm0.1}$ | $70.6_{\pm0.1}$ | $67.7_{\pm2.2}$ | $\underline{79.3}_{\pm0.3}$ | $\mathbf{80.4}_{\pm1.6}$ | |
| | Ogbn-arxiv [29] | 0.05% | $47.1_{\pm3.9}$ | $52.4_{\pm1.8}$ | $47.2_{\pm3.0}$ | $35.4_{\pm0.3}$ | $58.6_{\pm0.4}$ | $\underline{59.2}_{\pm1.1}$ | $\mathbf{60.8}_{\pm1.3}$ | $71.4_{\pm0.1}$ |
| | | 0.25% | $57.3_{\pm1.1}$ | $58.6_{\pm1.2}$ | $56.8_{\pm0.8}$ | $43.5_{\pm0.2}$ | $59.9_{\pm0.3}$ | $\underline{63.2}_{\pm0.3}$ | $\mathbf{65.8}_{\pm1.2}$ | |
| | | 0.50% | $60.0_{\pm0.9}$ | $60.4_{\pm0.8}$ | $60.3_{\pm0.4}$ | $50.4_{\pm0.1}$ | $59.5_{\pm0.3}$ | $\underline{64.0}_{\pm0.4}$ | $\mathbf{66.3}_{\pm0.7}$ | |
| | Flickr [100] | 0.10% | $41.8_{\pm2.0}$ | $42.5_{\pm1.8}$ | $42.0_{\pm0.7}$ | $41.9_{\pm0.2}$ | $46.3_{\pm0.2}$ | $\underline{46.5}_{\pm0.4}$ | $\mathbf{46.9}_{\pm0.1}$ | $47.2_{\pm0.1}$ |
| | | 0.50% | $44.0_{\pm0.4}$ | $43.9_{\pm0.9}$ | $43.2_{\pm0.1}$ | $44.5_{\pm0.1}$ | $45.9_{\pm0.1}$ | $\mathbf{47.1}_{\pm0.1}$ | $\underline{47.1}_{\pm0.3}$ | |
| | | 1.00% | $44.6_{\pm0.2}$ | $44.4_{\pm0.6}$ | $44.1_{\pm0.4}$ | $44.6_{\pm0.1}$ | $45.8_{\pm0.1}$ | $\underline{47.1}_{\pm0.1}$ | $\mathbf{47.1}_{\pm0.1}$ | |
| | Reddit [26] | 0.01% | $46.1_{\pm4.4}$ | $53.1_{\pm2.5}$ | $46.6_{\pm2.3}$ | $40.9_{\pm0.5}$ | $88.2_{\pm0.2}$ | $\underline{88.0}_{\pm1.8}$ | $\mathbf{90.5}_{\pm2.1}$ | $93.9_{\pm0.0}$ |
| | | 0.05% | $58.0_{\pm2.2}$ | $62.7_{\pm1.0}$ | $53.0_{\pm3.3}$ | $42.8_{\pm0.8}$ | $89.5_{\pm0.1}$ | $\underline{89.6}_{\pm0.7}$ | $\mathbf{91.8}_{\pm1.9}$ | |
| | | 0.50% | $66.3_{\pm1.9}$ | $71.0_{\pm1.6}$ | $58.5_{\pm2.1}$ | $47.4_{\pm0.9}$ | $\underline{90.5}_{\pm1.2}$ | $90.1_{\pm0.5}$ | $\mathbf{91.6}_{\pm1.8}$ | |
| AD | YelpChi [68] | 0.05% | $41.8_{\pm0.3}$ | $46.1_{\pm0.9}$ | $49.3_{\pm1.1}$ | $46.2_{\pm2.1}$ | $47.9_{\pm1.1}$ | $\underline{48.6}_{\pm3.7}$ | $\mathbf{56.2}_{\pm1.8}$ | $61.1_{\pm1.8}$ |
| | | 0.10% | $43.7_{\pm1.2}$ | $47.1_{\pm1.2}$ | $44.2_{\pm1.8}$ | $47.5_{\pm1.8}$ | $47.7_{\pm2.0}$ | $\underline{50.2}_{\pm2.1}$ | $\mathbf{58.1}_{\pm2.3}$ | |
| | | 0.20% | $45.9_{\pm2.2}$ | $46.4_{\pm0.8}$ | $47.5_{\pm0.4}$ | $49.1_{\pm1.2}$ | $49.7_{\pm2.0}$ | $\underline{50.1}_{\pm2.8}$ | $\mathbf{59.7}_{\pm1.8}$ | |
| | Amazon [102] | 0.02% | $76.2_{\pm0.8}$ | $74.1_{\pm0.9}$ | $73.4_{\pm2.1}$ | $75.2_{\pm2.9}$ | $74.1_{\pm1.9}$ | $\underline{77.9}_{\pm3.1}$ | $\mathbf{83.3}_{\pm2.6}$ | $89.5_{\pm0.9}$ |
| | | 0.20% | $76.4_{\pm1.6}$ | $76.5_{\pm2.3}$ | $74.2_{\pm1.1}$ | $76.8_{\pm1.0}$ | $\underline{78.2}_{\pm2.1}$ | $78.1_{\pm1.9}$ | $\mathbf{84.8}_{\pm1.7}$ | |
| | | 2.00% | $78.2_{\pm0.9}$ | $77.2_{\pm1.8}$ | $73.8_{\pm2.2}$ | $77.8_{\pm2.8}$ | $\underline{79.3}_{\pm3.1}$ | $79.2_{\pm2.0}$ | $\mathbf{86.9}_{\pm2.1}$ | |
| LP | Citeseer-L [97] | 0.90% | $56.8_{\pm0.8}$ | $63.1_{\pm1.3}$ | $78.3_{\pm2.6}$ | $76.9_{\pm0.8}$ | $81.5_{\pm2.5}$ | $\underline{83.4}_{\pm1.9}$ | $\mathbf{86.4}_{\pm1.6}$ | $96.8_{\pm1.8}$ |
| | | 1.80% | $61.4_{\pm1.8}$ | $63.3_{\pm1.8}$ | $79.1_{\pm1.8}$ | $77.4_{\pm1.7}$ | $83.4_{\pm1.6}$ | $\underline{83.8}_{\pm2.1}$ | $\mathbf{87.2}_{\pm2.1}$ | |
| | | 3.60% | $63.5_{\pm0.9}$ | $64.7_{\pm1.7}$ | $80.6_{\pm2.7}$ | $78.4_{\pm0.4}$ | $\underline{84.2}_{\pm2.1}$ | $83.1_{\pm2.7}$ | $\mathbf{87.1}_{\pm1.2}$ | |
| | DBLP [83] | 0.05% | $64.1_{\pm2.6}$ | $65.8_{\pm0.7}$ | $72.4_{\pm1.6}$ | $77.8_{\pm1.3}$ | $\underline{78.9}_{\pm2.1}$ | $77.2_{\pm2.1}$ | $\mathbf{81.3}_{\pm2.8}$ | $84.2_{\pm0.7}$ |
| | | 0.25% | $68.3_{\pm1.4}$ | $71.2_{\pm1.7}$ | $74.9_{\pm2.8}$ | $76.9_{\pm0.9}$ | $77.5_{\pm1.9}$ | $\underline{78.6}_{\pm1.2}$ | $\mathbf{82.1}_{\pm1.9}$ | |
| | | 0.50% | $68.7_{\pm2.6}$ | $74.3_{\pm0.8}$ | $75.2_{\pm0.6}$ | $76.8_{\pm2.1}$ | $77.6_{\pm2.1}$ | $\underline{79.9}_{\pm1.9}$ | $\mathbf{82.1}_{\pm1.8}$ | |

### 4.3 Training pipeline of SGDD

As illustrated in Fig. 1(d) and Fig. 3, we commence by introducing a novel graph structure learning paradigm termed "graphon approximation". This paradigm integrates both the feature $X'$ and auxiliary information $Z$ to generate the structure. Subsequently, the learned structure $A'$ is forced to be closer to the original graph structure $A$ in terms of Laplacian energy distribution.

Additionally, our proposed methodology, SGDD, implements a bi-loop optimization schema. Within this framework, we concurrently optimize the parameters $X'$ and $A'$. More specifically, the refinement of $X'$ is achieved through a gradient matching strategy, whereas the $A'$ is enhanced using the LED matching technique. During each step, the other component is frozen to ensure effective refinement.

Our overall training loss function can be summarized as $\mathcal{L} = \mathcal{L}_{feature} + \alpha\mathcal{L}_{structure} + \beta||\mathbf{A}||_2$, where the $||\mathbf{A}||_2$ is proposed as a sparsity regularization term and $\mathcal{L}_{feature}$ denotes the gradient matching strategy [37]. $\alpha$ and $\beta$ are trade-off parameters, we study their sensitiveness in the Sec. 5.3. The algorithm can be found in Appendix C.3.

## 5 Experiments

### 5.1 Datasets and implementation details

**Datasets.** We evaluate SGDD on five node classification datasets: Cora [41], Citeseer [41], Ogbn-arxiv [29], Flickr [100], Reddit [26], two anomaly detection datasets: YelpChi [68], Amazon [102], and two link prediction datasets: Citeseer-L [97], DBLP [83]. For the node classification and anomaly detection tasks, we follow the public settings [18] of train and test. To make a fair comparison, we also follow the previous setting [101, 83], we randomly split 80% nodes for training, 10% nodes for validation, and the remaining 10% for testing. To avoid data leakage, we only utilize 80% training samples for condensation. More details of each dataset can be found in Appendix C.1.

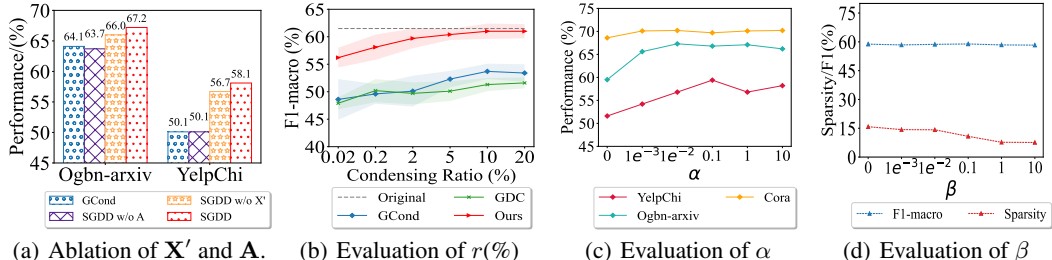

| (a) Ablation of $\mathbf{X}'$ and $\mathbf{A}$. | (b) Evaluation of $r(\%)$ | (c) Evaluation of $\alpha$ | (d) Evaluation of $\beta$ |

Figure 4: (a) Ablation of components in SGDD. (b) Evaluation of the scalability of SGDD. (c) and (d): the trade-off parameters analysis on $\alpha$ and $\beta$.

**Implementation details.** Without specific designation, **in the condense stage**, we adopt the 2-layer GCN with 128 hidden units as the backbone, and we adopt the settings on [94], which use 2-layer MLP to represent the structure generative model (*i.e.*, GEN). The learning rates for structure and feature are set to 0.001 (0.0001 for Ogbn-arxiv and Reddit) and 0.0001, respectively. We set $\alpha$ to 0.1, and $\beta$ to 0.1. **In the evaluation stage**, we train the same network for 1,000 epochs on the condensed graph with a learning rate of 0.001. Following the settings in [37], we repeat all experiments ten times and report average performance and variance. More details[4] can be found in Appendix C.2.

## 5.2 Comparison with state-of-the-art methods

We compare our proposed SGDD with six baselines: Random, which randomly selected nodes to form the original graph, corset methods (Herding [90] and K-Center [72]), graph coarsening methods (Corasening [30]), and the state-of-the-art graph condensation methods (GCond [37]). GDC is proposed as a baseline in [37], where cosine similarity is added as a constraint to generate the structure of the condensed graph. In Table 1, we present the performance metrics including accuracy, F1-macro, and AUC. For clarity and simplicity, percentages are represented without the % symbol, and variance values are provided. Based on the results, we have the following observations: 1) Our proposed SGDD achieves the highest results in most settings, which shows the superiority and generalization of our method. 2) On anomaly detection datasets, the improvements are more significant than other tasks, *i.e.*, improving GCond with 9.6% and 7.7% on YelpChi and Amazon datasets, which can be explained that our method captures the structure information from the original graph dataset more efficiently.

Table 2: Results of cross-architecture setting, we test condensed graphs in APPNP, Cheby, GCN, GraphSAGE, and SGC. Avg. and Std. : the average performance and the standard deviation of the results, the $\Delta(\%)$ denotes the improvements upon the GDC. We mark the best performance by **bold**.

| Datasets | Methods | Architectures | | | | | | | Statistics | | |
| | | MLP [11] | GAT [84] | APPNP [42] | Cheby [13] | GCN [41] | SAGE [26] | SGC [92] | Avg. | Std. | $\Delta(\%)$ |
|---|---|---|---|---|---|---|---|---|---|---|---|
| Reddit [26] | GDC | 50.3 | 54.8 | 81.2 | 77.5 | 89.5 | 89.7 | 90.5 | 76.2 | 16.9 | - |
| | GCond | 42.5 | 60.2 | 87.8 | 75.5 | 89.4 | 89.1 | 89.6 | 76.3 | 18.5 | ↑ 0.1 |
| | Ours | 56.1 | 74.4 | 89.2 | 78.4 | 89.4 | 89.4 | 89.4 | **80.9** | **12.6** | ↑ **4.7** |
| Cora [41] | GDC | 67.2 | 64.2 | 67.1 | 67.7 | 67.9 | 66.2 | 72.8 | 67.6 | 2.6 | - |
| | GCond | 73.1 | 66.2 | 78.5 | 76 | 80.1 | 78.2 | 79.3 | 75.9 | 4.9 | ↑ 8.3 |
| | Ours | 76.7 | 75.8 | 78.4 | 78.5 | 79.8 | 80.4 | 78.5 | **78.3** | **1.6** | ↑ **10.7** |
| DBLP [83] | GDC | 74.4 | 76.8 | 77.4 | 76.7 | 78.9 | 74.8 | 78.4 | 76.8 | 1.7 | - |
| | GCond | 75.3 | 77.6 | 78.9 | 76.1 | 79.6 | 77.4 | 79.9 | 77.8 | 1.7 | ↑ 1.1 |
| | Ours | 78.4 | 79.6 | 80.1 | 80.6 | 82.1 | 80.7 | 81.4 | **80.4** | **1.2** | ↑ **3.6** |
| YelpChi [68] | GDC | 30.7 | 36.4 | 43.7 | 41.5 | 49.6 | 47.4 | 50.1 | 42.8 | 7.2 | - |
| | GCond | 48.9 | 31.8 | 46.7 | 48.6 | 50.1 | 42.5 | 48.7 | 45.3 | 6.5 | ↑ 2.6 |
| | Ours | 54.2 | 56.4 | 58.2 | 56.8 | 59.7 | 54.1 | 56.7 | **56.6** | **2.0** | ↑ **13.8** |

## 5.3 Ablation Study

**Cross-architecture generalization analysis.** To evaluate the generalization ability of SGDD on unseen architectures, we conduct experiments that train on the condensed graph with different

---

[4]The code is available in https://github.com/RingBDStack/SGDD, based on PyTorch[65] and MindSpore[31].

Table 3: Comparison of the cross-architecture generalization performance between GCond and SGDD on Ogbn-arxiv. **Bold entries** are the best results. ↑/↓: our method show increase or decrease performance.

| C\T | APPNP GCond / SGDD | Cheby GCond / SGDD | GCN GCond / SGDD | SAGE GCond / SGDD | SGC GCond / SGDD |
|---|---|---|---|---|---|
| APPNP | **60.3** / 60.2↓ | 51.8 / **53.2**↑ | 59.9 / **62.4**↑ | 59.0 / **60.2**↑ | **61.2** / 60.4↓ |
| Cheby | 57.4 / **58.5**↑ | 53.5 / **55.8**↑ | 57.4 / **65.3**↑ | **57.1** / 57.0↓ | 58.2 / **60.2**↑ |
| GCN | 59.3 / **60.1**↑ | 51.8 / **53.7**↑ | 60.3 / **64.2**↑ | 60.2 / **61.2**↑ | 59.2 / **59.8**↑ |
| SAGE | 57.6 / **58.9**↑ | 53.9 / **53.8**↓ | 58.1 / **63.8**↑ | 57.8 / **61.8**↑ | 59.0 / **61.1**↑ |
| SGC | 59.7 / **60.0**↑ | 49.5 / **52.3**↑ | 59.2 / **62.2**↑ | 58.9 / **62.9**↑ | 60.5 / **61.5**↑ |

Table 4: Neural Architecture Search. Methods are compared in validation accuracy correlation and test accuracy on obtained architecture.

| Dataset | Pearson Correlation / Performance (%) | | | Whole Per. (%) |
|---|---|---|---|---|
| | Random | GCond | SGDD | |
| Ogbn-arxiv | 0.63 / 71.1 | 0.64 / 71.2 | **0.67 / 71.6** | 71.9 |
| YelpChi | 0.43 / 56.7 | 0.48 / 58.4 | **0.56 / 60.6** | 61.1 |
| DBLP | 0.58 / 81.2 | 0.62 / 83.8 | **0.68 / 84.0** | 84.2 |

architectures and report their performances in Tab. 2. Here, the condensed graph is obtained by optimizing SGDD with SGC [92]. We test its cross-architecture generalization performances on 2-layer-MLP [11], GAT [84], APPNP [42], Cheby [13], GCN [41], and SAGE [26].

To better understand, we also show several statistics metrics, including Avg., Std., and $\Delta$. SGDD and GCond improves GDC significantly, which indicates there exists a large difference between graph dataset condensation and vision dataset distillation, *i.e.*, the structure information should be specially considered. Compared to GCond, the improvement of our method is up to 11.3%, demonstrating the effectiveness of broadcasting the original structure to the condensed graph structure generation. More experiments on other datasets can be found in Appendix C.6.

**Versatility of SGDD.** Following the setting of GCond [37], we also study whether our proposed SGDD is robust on various architectures. We first condense the Ogbn-arxiv graph dataset with five architectures, including APPNP [42], Cheby [13], GCN [41], SAGE [26], and SGC [92], respectively. Then, we evaluate these condensed graphs on the above five architectures and report their performances in Tab. 3. The experiment results show that SGDD achieves non-trivial improvements than GCond in most cases, which demonstrates the strong versatility of our method.

**Evaluation on neural architecture search.** Similar to vision dataset distillation, graph dataset distillation is also expected to reduce the high cost of neural architecture search (NAS). In order to make a fair comparison, we follow the experimental setting in [37]: searching architectures on condensed Obgn-arxiv, YelpChi, and DBLP datasets with 0.25%, 0.2%, and 0.5% condensing ratios. We report Pearson correlation [45] and performance of random, GCond, and SGDD in Tab. 4. Our SGDD consistently achieves the highest Pearson correlations as well as performances, which indicates the architectures searched by our method are efficient for the whole graph dataset training.

**Evaluation of components in SGDD.** To explore the effect of the conditions (mentioned in Sec. 4.1) in the condensed graph generation, we design the ablation study of $\mathbf{X}'$ and $\mathbf{A}$. As shown in Fig 4(a), $\mathbf{X}'$ and $\mathbf{A}$ are complementary with each other. SGDD w/o $\mathbf{A}$ performs poorly on Obgn-arxiv and YelpChi datasets, which demonstrates the effect of original graph structure information. Jointly using $\mathbf{X}'$ and $\mathbf{A}$ achieves the highest performances on both datasets, improves GCond with 3.1% on Ogbn-arxiv, and 8.0% on YelpChi.

**Evaluation of the scalability of SGDD.** To investigate the scalability of SGDD, we evaluate the SGDD on various condensing ratios with $r \in \{0.02, 0.2, 2, 5, 10, 20\}$. As shown in Fig. 4(b), the performance of our method continuously increase as the condensing ratio rises, which indicates the strong scalability of our method. GCond obtains marginal improvements than GDC at all ratios while our SGDD outperforms them significantly. More important, SGDD achieves lossless performance as training on the original graph data when the condensing ratio is 10%.

**Exploring the sensitivity of $\alpha$ and $\beta$.** We conduct experiments to test such two parameters sensitivity on YelpChi, Cora, and Ogbn-arxiv. As shown in Fig. 4(c), we empirically find that the performance of our SGDD is not sensitive to the $\alpha$. Specifically, compared to the case that $\alpha$ is zero, we have a significant improvement, which proves the effectiveness of our method. Another finding is that the $\alpha$ should be set higher on anomaly detection than on node classification tasks. It could be explained by the original graph structure information being more important on the complex task (such as anomaly detection). We define $\beta$ as a regularization coefficient to control the sparsity of the condensed graph. As shown in Fig. 4(d), we evaluate the $\beta$ from 0 to 10 on the YelpChi dataset. The results illustrate that the performance is not sensitive with $\beta$ and achieve the highest result (F1-macro) when $\beta$ is set to our default value ($\beta = 0.1$). More experiments can be found in Appendix C.5.

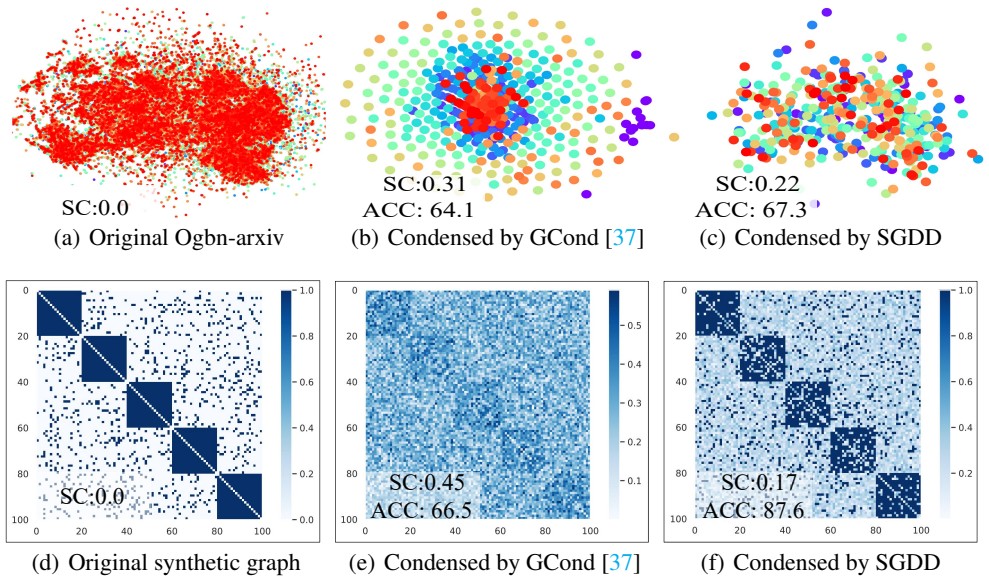

Figure 5: Visualizations of the real dataset (a), synthetic dataset (d), and the corresponding condensed graph obtained by GCond (b, e) and SGDD (c, f). The condensing ratio $r$ is set to 0.5%.

## 5.4 Visualizations

To better understand the effectiveness of our SGDD, we visualize the condensed graphs of GCond and SGDD that are synthesized from real and synthetic graph datasets. For the synthetic graph dataset, we use the Stochastic Block Model (SBM) [28] to synthesize graphs with 5 community (Fig. 5(d)). As shown in Fig. 5, one can find that our method consistently achieves better performances and $SC$. SGDD reduces 29.0% (comparing 5(b) and 5(c)) and 62.2% (comparing 5(e) and 5(f)) $SC$ on Ogbn-arxiv and synthetic datasets. Visually, the condensed graphs of our method preserve the original graph structure information obviously better than GCond (see the second row of Fig. 5), which proves SGDD is a powerful graph dataset distillation method.

## 6 Conclusion

We present SGDD, a novel framework for graph dataset distillation via broadcasting the original structure information to the generation of the synthetic one. SGDD shows its robustness on various tasks and datasets, achieving state-of-the-art results on YelpChi, Amazon, Ogbn-arxiv, and DBLP. SGDD reduces the scale of the Yelpchi dataset by 1,000 times while maintaining 98.6% as training on the original data. We provide sufficient experiments and theoretical analysis in this paper and hope it can help the following research in this area. **Limitations and future work:** Although broadcasting the original information to the generated graph shows remarkable success, some informative properties (*e.g.*, the heterogeneity) may lose during the current condense process, which results in sub-optimal performance in the downstream tasks. We are going to explore a more general method in the future.

## Acknowledgement

The corresponding author is Jianxin Li. This work was supported by the NSFC through grant No.62225202 and No.62302023. This research is supported by the National Research Foundation, Singapore under its AI Singapore Programme (AISG Award No: AISG2-PhD-2021-08-008). Yang You's research group is being sponsored by NUS startup grant (Presidential Young Professorship), Singapore MOE Tier-1 grant, ByteDance grant, ARCTIC grant, SMI grant, and Alibaba grant. We gratefully acknowledge the support of MindSpore, CANN(Compute Architecture for Neural Networks) and Ascend AI Processor used for this research.

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

# A   More Preliminary and Related Work

## A.1   More preliminary

**Graphon.** A graphon [43, 34, 54] is a bounded, symmetric, and Lebesgue measurable function, denote as $W : \Omega^2 \to [0, 1]$, where $\Omega$ is a probability measure space. By randomly selecting points $\{v_0, v_1, \ldots, v_n\}$ from $\Omega$, we can create a graph of any size by connecting each point with an edge weight $e_{ij} = W(v_i, v_j)$. Formally, we can follow the random sampling process to get arbitrarily sized ($n$) graphs :

$$v_i \sim \mathrm{Uniform}(\Omega), \text{ for } i = 1, \cdots, n,$$
$$e_{ij} \sim \mathrm{Bernoulli}(W(v_i, v_j)), \text{ for } i, j = 1, \cdots, n. \tag{10}$$

Conventionally, we use a deterministic setting to select the node, *e.g.*, the fixed grid $v_i = \frac{i-1}{n}$. Graphs derived from the same graphon share several important properties for statistical analysis: such as density [21], clustering coefficient [70], and degree distribution [71], which motivate us to broadcast the original graph structure information to the condensed graph via the graphon approximating.

## A.2   More related work

**Graph structure learning.** Graph structure learning [19, 35, 8, 108, 55] also aims at jointly optimizing the graph structure and the corresponding node features. Most of these methods learn the new structure based on the certain constraints (*e.g.*, low-rank [91], sparsity [8], feature smoothness [35], and homophily [108]). However, these methods are unable to learn a new structure with a significantly reduced node size. In contrast, our method can synthesize a smaller graph structure while simultaneously constructing new node features. The obtained small but informative graph can reduce the training cost in the downstream tasks.

# B   Proofs

## B.1   Proof of Proposition 1

*Proof.* First, the gradient matching objective in GCond [37] can be shown as:

$$\mathrm{Match}(\nabla_{\boldsymbol{\theta}} \mathcal{L}_{cls}(\mathrm{GNN}_{\boldsymbol{\theta}}(\mathbf{A}, \mathbf{X}), \mathbf{Y})), \quad \nabla_{\boldsymbol{\theta}} \mathcal{L}_{cls}(\mathrm{GNN}_{\boldsymbol{\theta}}(\mathbf{A}', \mathbf{X}'), \mathbf{Y}')), \tag{11}$$

where $\mathrm{Match}(\cdot)$ is the matching function that measures the distance of the two gradients. $\theta$ represents the parameters of the GNN, the $\nabla$ denotes the gradient in the backpropagation process, and the $\mathcal{L}_{cls}$ represents the loss function that is used in the supervised tasks, *i.e.*, cross-entropy loss. Following the discussion in the [4, 1, 82], the GNN can be viewed as a bandpass filter in the spectral domain. Expanding the GNN in the spectral domain, we can rewrite the objective as:

$$\left\| \int_{i=0}^{N} \mathbf{U_A} \operatorname{diag}(\mathrm{T}_i(\boldsymbol{\lambda_A})) \mathbf{U_A}^{\top} \mathbf{X} - \int_{j=1}^{N'} \mathbf{U_{A'}} \operatorname{diag}(\mathrm{T}_j(\boldsymbol{\lambda_{A'}})) \mathbf{U_{A'}}^{\top} \mathbf{X'} \right\| \leq \epsilon, \tag{12}$$

where the $\mathrm{T}(\boldsymbol{\lambda})$ denotes the frequency response of the given GNN, the $\boldsymbol{\lambda}$ and the $\mathbf{U}$ is the eigenvalue and eigenvector of the corresponding graph, respectively. Note we drop the $\nabla$, the $\mathcal{L}_{cls}$, and the $\mathbf{Y}$ ($\mathbf{Y}'$) terms because they can be viewed as the intermediate process [105, 107] in approximating to Eq.(12). We use the MSE [37] as the matching function $\mathrm{Match}(\cdot)$ for simplicity.

To simplify $\mathrm{T}(\boldsymbol{\lambda})$, without loss of generality, we assume the bandwidth of the specific GNN's frequency response is (a, b), which intuitively indicates that only the signal of frequency in (a, b) can pass through the filter on the graph. Then the Eq. (12) can be written as:

$$\left\| \int_{i=0}^{N} \int_{k=a}^{b} \mathbf{U_A} \operatorname{diag}(\boldsymbol{\lambda_A}^k) \mathbf{U_A}^{\top} \mathbf{X} - \int_{j=1}^{N'} \int_{k=a}^{b} \mathbf{U_{A'}} \operatorname{diag}(\boldsymbol{\lambda_{A'}}^k) \mathbf{U_{A'}}^{\top} \mathbf{X'} \right\| \leq \epsilon, \tag{13}$$

we further combine the first integration, the objective can thus be summarized as:

$$\left\| \int_{k=a}^{b} \left( \mathbf{U_A} \operatorname{diag}(\boldsymbol{\lambda_A}^k) \mathbf{U_A}^{\top} \mathbf{X} - \mathbf{U_A} \operatorname{diag}(\boldsymbol{\lambda_{A'}}^k) \mathbf{U_{A'}}^{\top} \mathbf{X'} \right) \right\| \leq \epsilon. \tag{14}$$

Then following the transformation in the spectral domain [1], we can briefly summarized:

$$\int_{i=a}^{b} ||\bar{x}_i^2 - \bar{x}_i^{'2}|| \leq \epsilon. \tag{15}$$

Take the Eq. (15) into the $\left\|\eta^{\mathcal{G}} - \eta^{\mathcal{S}}\right\|$:

$$||\eta^{\mathcal{G}} - \eta^{\mathcal{S}}|| = || \sum_{i=1}^{N} \bar{x}_i^2 - \sum_{i=j}^{N'} \bar{x}_i^2 ||$$

$$\geq \epsilon + || \sum_{i=1}^{a} \bar{x}_i^2 - \sum_{j=1}^{a} \bar{x}_j^2 || + || \sum_{i=b}^{N} \bar{x}_i^2 - \sum_{j=b}^{N'} \bar{x}_j^2 || \tag{16}$$

$$\geq \epsilon + \sum_{i=1}^{a} \left\|\bar{x}_i^2 - \bar{x}_i'^2\right\| + \sum_{i=b}^{N'} \left\|\bar{x}_i^2 - \bar{x}_i'^2\right\|.$$

Note the second inequality is under the assumption: intuitively, the overall distance of $\eta^{\mathcal{G}}$ and $\eta^{\mathcal{S}}$ should be larger than the condition that two graphs have the same size (*i.e.*, $N = N'$). $\qquad \square$

## B.2   Proof of Proposition 2

To prove Proposition 2, we first introduce the following notations and theorems in graphon theory.

The cut norm [20, 54] is defined as:

$$\|W\|_{\square} := \sup_{\mathcal{X}, \mathcal{Y} \subset \Omega} \left| \int_{\mathcal{X} \times \mathcal{Y}} W(x, y) \mathrm{d}x\mathrm{d}y \right|, \tag{17}$$

where the $\Omega$ is the probability space of a graphon $W$, the supremum is taken over all measurable subsets $\mathcal{X}$ and $\mathcal{Y}$ [95]. Then the cut distance between $W_1, W_2$ [54] can be defined as:

$$\delta_{\square}(W_1, W_2) := \inf_{\phi \in \mathcal{S}_{\Omega}} \left\|W_1 - W_2^{\phi}\right\|_{\square}, \tag{18}$$

where the $\mathcal{S}_{\Omega}$ is the set of measure-preserving mappings from $\Omega$ to $\Omega$. When two graphons $W_1, W_2$ have $\delta_{\square}(W_1, W_2) = 0$, they can be seen equivalent, denoted as $W_1 \cong W_2$.

To effectively approximate $W$ in the real world graphs, works [54, 33, 17] introduce an approximation named step function. Let $\mathcal{P} = (\mathcal{P}_1, \dots, \mathcal{P}_K)$ be a partition of $\Omega$ into $K$ measurable sets. The step function $W_{\mathcal{P}} : \Omega^2 \mapsto [0, 1]$ is defined as:

$$W_{\mathcal{P}}(x, y) = \sum_{k, k'=1}^{K} w_{kk'} \mathbb{1}_{\mathcal{P}_k \times \mathcal{P}_{k'}}(x, y), \tag{19}$$

where the $w_{kk'} \in [0, 1]$ and the indicator function $\mathbb{1}_{\mathcal{P}_k \times \mathcal{P}_{k'}}(x, y)$ is 1 if $(x, y) \in \mathcal{P}_k \times \mathcal{P}_{k'}$, or it is 0.

To explore the relationship between step function and the $W$, we introduce the Weak Regularity Lemma as follows.

**Theorem 1** (Weak Regularity Lemma [54]). *For every graphon $W \in \mathcal{W}$ and $K \leq 1$, there always exists a step function $W_{\mathcal{P}}$ with $|\mathcal{P}| = K$ steps such that*

$$\|W - W_{\mathcal{P}}\|_{\square} \leq \frac{2}{\sqrt{\log K}} \|W\|_{L_2}. \tag{20}$$

We can further obtain the corollary that $\delta_{\square}(W, W_{\mathcal{P}}) \leq \frac{2}{\sqrt{\log K}} \|W\|_{L_2}$ as the $\delta_{\square}(W, W_{\mathcal{P}}) \leq \|W - W_{\mathcal{P}}\|_{\square}$. Intuitively, we can use any step function to approximate the ideal $W$ of real graphs.

Then to investigate the properties of the graphon in the spectral domain, following [71, 85, 59, 69], we have the conclusion that:

$$||S_W - S_{W_{\mathcal{P}}}|| \leq \|W - W_{\mathcal{P}}\|_{\square}, \tag{21}$$

where the $S$ denotes the signal spectrum of the graphs [85] (*i.e.*, the post-Graph-Fourier-Transform $\mathbf{U}^{\top}\mathbf{X}$), Intuitively, the Eq. (21) shows that when the step function $W_{\mathcal{P}}$ is approximated to the $W$, they have similar signal spectrum.

*Proof.* By combining the Theorem 1 and Eq.(21), the Proposition 2 replace the $W_{\mathcal{P}}$ with the $W'_{\mathbf{A}}$, that's because we use the generative model (*i.e.*, $\text{GEN}(\cdot)$) to approximate the step function $W_{\mathcal{P}}$. As we aim to learn the graphon of the original structure $\mathbf{A}$, the graphon $W$ can be rewrite as $W_{\mathbf{A}}$,

$$\left|\left| S_{W_{\mathbf{A}}} - S_{W'_{\mathbf{A}}} \right|\right| \le \left\| W'_{\mathbf{A}} - W_{\mathbf{A}} \right\|_{\square}. \tag{22}$$

We notice that the $S$ here is related to the Laplacian energy distribution (LED), where the LED represents the probability distribution of the $S$, then we use the $\sum_{i=1}^{N} \bar{x}_i$ and $\sum_{j=1}^{N'} \bar{x}_j$ to represent the summation of the original signal spectrum and the condensed one, respectively, we have:

$$\begin{aligned} \|\eta^{\mathcal{G}} - \eta^{\mathcal{S}}\| &= \|\frac{S_{W_{\mathbf{A}}}}{\sum_{i=1}^{N} \bar{x}_i} - \frac{S_{W_{\mathbf{A}'}}}{\sum_{j=1}^{N'} \bar{x}_j}\| \\ &\le \max(\frac{1}{\sum_{i=1}^{N} \bar{x}_i}, \frac{1}{\sum_{j=1}^{N'} \bar{x}_j}) \left|\left| S_{W_{\mathbf{A}}} - S_{W'_{\mathbf{A}}} \right|\right| \\ &\le \|W'_{\mathbf{A}} - W_{\mathbf{A}}\|_{\square}, \end{aligned} \tag{23}$$

where in the second inequality, we drop the $\max(\cdot)$ term since it always lower than 1. As the $W_{\mathbf{A}'}$ here represents the graphon of the synthetic $\mathbf{A}'$, we can use the $\mathbf{A}'$ directly in the $\delta_{\square}$ to form a compact upper bound.

$$\|\eta^{\mathcal{G}} - \eta^{\mathcal{S}}\| \le \delta_{\square}(\mathbf{A}', W_{\mathbf{A}}). \tag{24}$$

$\square$

Note that minimizing the upper bound has been proven to be equivalent to minimizing the optimal transport distance between the two graphs [95].

## B.3  Time complexity analysis and running time

**Time complexity.** For simplicity, let the number of MLP layers in $\text{GEN}(\cdot)$ be $L$, and all the hidden units are $d$. *In the forward process*, we have three steps: first, we calculate the $\mathbf{A}'$ by the $\text{GEN}(\cdot)$, which have the complexity of $\mathcal{O}(N'^2 d^2)$. Second, the forward process of GNN on the original graph has a complexity of $\mathcal{O}(m^L N d^2)$, where the $m$ denotes the sampled size per node in training. Third, the complexity of training on the condensed graph is $\mathcal{O}(LN'd)$. *In the backward process*, the complexity of gradient matching strategy (*i.e.*, $\mathcal{L}_{feature}$) is $\mathcal{O}(|\theta||\mathbf{X}'|)$ [37]. For the structure optimization term (*i.e.*, $\mathcal{L}_{structure}$), the complexity is $\mathcal{O}(N'^2 k + NN'^2)$. The overall complexity of SGDD can be represented as $\mathcal{O}(N'^2 d^2) + \mathcal{O}(m^L N d^2) + \mathcal{O}(LN'd) + \mathcal{O}(|\theta||X'|) + \mathcal{O}(N'^2 k + N'^2 N)$. Note $N' \ll N$, we can drop the terms that only involve $N'$ and constants (*e.g.*, the number of $L$ and $m$). The final complexity can be simplified as $\mathcal{O}(m^L N d^2) + \mathcal{O}(N'^2 N)$, thus the complexity of SGDD still be linear to the number of nodes in the original graph.

**Running time.** We report the running time of the SGDD in the two datasets: Ogbn-arxiv, and YelpChi. We vary the condensing ratio $r$ in the range of {0.05%, 0.25%, 0.50%} for Ogbn-arxiv and {0.05%, 0.10%, 0.20%} for YelpChi. All experiments are conducted five times on one single A100-SXM4 GPU. We also compare our results to those obtained using GCond [37] under the same settings. As shown in the Tab. 5, our approach achieves a similar running time to GCond when the condensing ratio was low (*i.e.*, $r = 0.05\%$ for both datasets), and is 10% faster when the condensing ratio increased. The difference can be explained by the efficient generative model we employed for generating structure, which prevents the consumption of time-complex operations such as calculating the pair-wised feature similarity ($\mathcal{O}(N'^2)$).

Table 5: Runing time on Ogbn-arxiv and YelpChi for 50 epochs.

| Dataset | $r$ | GCond | SGDD | Dataset | $r$ | GCond | SGDD |
|---|---|---|---|---|---|---|---|
| | 0.05% | 315±1.8s | 308±1.6s | | 0.05% | 67±2.6s | 47±2.8s |
| Ogbn-arxiv | 0.25% | 413±2.6s | 374±3.2s | YelpChi | 0.10% | 96±2.8s | 74±1.7s |
| | 0.50% | 527±2.7s | 467±2.1s | | 0.20% | 110±0.8s | 93±2.6s |

# C Experimental Details and More Experiments

## C.1 Dataset statistics

We evaluate the proposed SGDD on nine datasets, including five node classification datasets: Cora [41], Citeseer [41], Ogbn-arxiv [29], Flickr [100], and Reddit [26]; two anomaly detection datasets: YelpChi [68] and Amazon [102]; two link prediction datasets Citeseer-L [97] and DBLP [83]. We report the dataset statistics in Tab. 6.

Table 6: Dataset statics, including five node classification datasets, two anomaly detection datasets, and two link prediction datasets.

|     | **Datasets** | **#Nodes** | **#Edges** | **#Classes** | **#Features** |
|-----|--------------|-----------|-----------|-------------|--------------|
| **ND** | Cora [41] | 2,708 | 5,429 | 7 | 1,433 |
|     | Citeseer [41] | 3,327 | 4,732 | 6 | 3,703 |
|     | Ogbn-arxiv [29] | 169,343 | 1,166,243 | 40 | 128 |
|     | Flickr [100] | 89,250 | 899,756 | 7 | 500 |
|     | Reddit [26] | 232,965 | 57,307,946 | 210 | 602 |
| **AD** | YelpChi [68] | 45,954 | 3,846,979 | 2 | 32 |
|     | Amazon [102] | 11,944 | 4,398,392 | 2 | 25 |
| **LP** | Citeseer-L [41] | 3,327 | 4,732 | 2 | 3,703 |
|     | DBLP [83] | 26,128 | 105,734 | 2 | 4,057 |

Citeseer-L: We use the Citeseer in the link prediction setting, named Citeseer-L. We randomly sample 80% nodes in training, 10% nodes in validation, and the remaining 10% nodes for testing. The classes here denote "have edge" and "do not have edge".

DBLP: We treat the original graph as a homogeneous graph here.

## C.2 Implementation details

**Structure in GDC.** We utilize the DC [107] as our baseline, and to incorporate the structure information, we add the constraint to produce a graph structure, named Graph DC (GDC). Specifically, we use the cosine similarity function [8] (formally, $\mathbf{A}'_{ij} = \cos(\mathbf{X}'_i, \mathbf{X}'_j)$) to generate structure, where $\mathbf{X}'_i$ and $\mathbf{X}'_j$ are the learned features obtained through the vanilla gradient matching strategy.

$\text{GEN}(\cdot)$ **in SGDD.** We introduce the $\text{GEN}(\cdot)$ as our generative model in Sec. 4.1. Here, we show the implementation details of this module.

To start, we aim to find a method to broadcast the original graph structure $\mathbf{A}$ to condensed graph $\mathbf{A}'$. Motivated by that all graphs with the same graphon $W$ will exhibit similar properties, we can simply calculate the graphon $W$ of $\mathbf{A}$ and leverage it in the condensing process. Nevertheless, directly calculating $W$ of $\mathbf{A}$ is not feasible since conventional graphon learning methods should summarize graphon from a set of graphs [21, 33, 64] (We only have one graph $\mathbf{A}$). Recent advancements of IGNR [94] demonstrate the potential of utilizing the generative approach to approximate the $W$. Specifically, given that the graphon $W$ is defined as $\Omega^2 \to [0, 1]$ (as described in Appendix A.1), we can similarly construct the function $f$ as follows.

$$f : \mathbb{R}^2 \to [0, 1], \tag{25}$$

the continuous space $\Omega^2$ is defined by $\mathbb{R}^2$. For computational convenience, the input space can further be limited to $[0, 1]^2$ [94], then the Eq. (25) is transformed to sample points to reconstruct data, following IGNR[94], we use SIREN[76] as $f$,

$$\begin{aligned}
\mathbf{h}_0 &= \text{PostionalEncoding}(\mathcal{Z}(N')), \\
\mathbf{h}_i &= \text{Sin}(\mathbf{W}_i(\mathbf{h}_{i-1}) + \mathbf{b}_i), \ i = 1, \cdots, l-1, \\
\mathbf{h}_l &= \text{Sigmoid}(\mathbf{W}_l \mathbf{h}_{l-1} + \mathbf{b}_l),
\end{aligned} \tag{26}$$

where the $\mathcal{Z}(N') \in \mathbb{R}^{N' \times N'}$ is a random noise that plays as the coordinates, and the learnable weights $\Phi = \{\mathbf{W}_i \in \mathbb{R}^{l_i \times l_{i+1}}, \mathbf{b}_i \in \mathbb{R}^{l_i}, \ \text{for } i = 1, \cdots, l \}$ map the pair of points to the edge probability. $\text{GEN}(\mathcal{Z}(N'); \Phi) \in \mathbb{R}^{N' \times N'}$ is equal to represent the adjacency matrix $\mathbf{A}' \in \mathbb{R}^{N' \times N'}$

after transformation, where each entry represents a probability that each node pair should be connected. To incorporate the node information into the structure generation, we adopt them as conditional information that leads the generation process. We can then rewrite Eq. (25) to:

$$f : \mathbb{R}^d \times \mathbb{R}^2 \to [0, 1], \tag{27}$$

where the $\mathbb{R}^d$ here is to present the conditional information, thus the learned model considers both the node's coordinates message along with the node's specific information. The basic implements can be:

$$\mathbf{h}_i = \text{MLP}_i(\mathbf{X}' \oplus \mathbf{Y}') \oplus \text{Sin}(\mathbf{W}_i \mathbf{h}_{i-1} + \mathbf{b}_i), \quad i = 1, \cdots, l, \tag{28}$$

where the $\oplus$ denotes the concatenate operation, the $\mathbf{Y}'$ is treated as a one-hot vector for dimensionality fit through a multilayer perceptron (MLP). This method allows for the incorporation of significant node information into the resulting synthetic graph.

**Condensation stage.** For GCond [37], we use the 2-layer SGC [92] to serve as the condensing architecture with 256 units, and tune the number of epochs in a range of {400, 500, 6000, 1000, 2000}. For GDC [107, 105], we tune the number of hidden layers in the range of {1, 2, 3} and the number of hidden units in the range of {128, 256}. For SGDD, we use the GCN [40] as the default condensing architecture and tune the number of hidden layers in a range of {1, 2, 3}. We further tune the number of epochs in a range {400, 500, 600, 1000, 2000}, and tune the learning rate in a range of {0.1, 0.01, 0.001, 0.0001}.

**Evaluation stage.** We set the training epoch to 1000 with an early stopping strategy for evaluating GNNs and set the dropout rate to 0 with the learning rate of 0.1.

**Configurations.** We conduct all experiments with:

- Operating System: Ubuntu 20.04 LTS.
- CPU: Intel(R) Xeon(R) Platinum 8358 CPU@2.60GHz with 1TB DDR4 of Memory.
- GPU: NVIDIA Tesla A100 SMX4 with 40GB of Memory.
- Software: CUDA 10.1, Python 3.8.12, PyTorch [65] 1.7.0.

### C.3 Objective loss function and training algorithm

In this subsection, we present the objective function and provide a detailed training algorithm. Our objective is to jointly learn $\mathbf{X}'$ and $\mathbf{A}'$. We follow GCond[37] to optimize $\mathbf{X}'$ as a free parameter using the gradient matching strategy[104], the loss can be expressed by Eq. (29),

$$\mathcal{L}_{feature} = \text{Match}(\nabla_{\boldsymbol{\theta}} \mathcal{L}_{cls} (\text{GNN}_{\boldsymbol{\theta}}(\mathbf{A}, \mathbf{X}), \mathbf{Y})), \quad \nabla_{\boldsymbol{\theta}} \mathcal{L}_{cls} (\text{GNN}_{\boldsymbol{\theta}}(\mathbf{A}', \mathbf{X}'), \mathbf{Y}')). \tag{29}$$

Here, $\text{Match}(\cdot)$ is the matching function that measures the distance of the two gradients, we use the MSE [37] in practice. $\theta$ represents the parameters of the specific backbone GNN, the $\nabla$ denotes the gradient in the backpropagation process, and the $\mathcal{L}_{cls}$ represents the loss function that is used in the supervised tasks, *i.e.*, cross-entropy loss. Our objective loss function can be written as:

$$\mathcal{L} = \mathcal{L}_{feature} + \alpha \, \mathcal{L}_{strcuture} + \beta \, ||\mathbf{A}'||_2, \tag{30}$$

where the $\alpha$ controls the contribution of the $L_{strcuture}$ term. To model the low-rank properties of real-world graphs, we use the $||\mathbf{A}'||_2$ as a regulary to control the sparsification of $\mathbf{A}'$ with $\beta$.

We summarize our pipeline in Algorithm. 1.

### C.4 Stochastic Block Model experments setting

In Sec. 5.4, we generate a synthetic graph dataset with different community structures using the Stochastic Block Model (SBM) $(N, C, p, q)$ [28]. Here, we show the parameters settings, we set the number of nodes $N$ to 100, while setting the number of communities $C$ to 5. The parameter $p$ represents the edge probability within the same community and $q$ represents the edge probability between communities, we set them to 0.8 and 0.1 in practice, respectively.

**Algorithm 1:** SGDD for Graph Condensation

---
1  **Input:** Training data $\mathcal{G} = (\mathbf{A}, \mathbf{X}, \mathbf{Y})$, pre-defined condensed labels $\mathbf{Y}'$
2  Initialize GEN as the structure learning model
3  Initialize $\mathbf{X}'$ by randomly seleclet node feature from each class
4  **for** $k = 0, \ldots, K - 1$ **do**
5  $\quad$ Randomly initialize GNN$_\theta$
6  $\quad$ **for** $t = 0, \ldots, T - 1$ **do**
7  $\quad\quad$ $D' = 0$
8  $\quad\quad$ **for** $c = 0, \ldots, C - 1$ **do**
9  $\quad\quad\quad$ Initialize $\mathcal{Z}(N')$
10 $\quad\quad\quad$ Compute $\mathbf{A}' = \text{GEN}(\mathcal{Z}(N') \oplus \mathbf{X}' \oplus \mathbf{Y}'; \Phi)$ then $\mathcal{S} = \{\mathbf{A}', \mathbf{X}', \mathbf{Y}'\}$
11 $\quad\quad\quad$ Sample $(\mathbf{A}_c, \mathbf{X}_c, \mathbf{Y}_c) \sim \mathcal{G}$ and $(\mathbf{A}'_c, \mathbf{X}'_c, \mathbf{Y}'_c) \sim \mathcal{S}$
12 $\quad\quad\quad$ Compute $\mathcal{L}_{structure}$ $\qquad\qquad\qquad$ ▷ detailed in Eq. (9)
13 $\quad\quad\quad$ Compute $\mathcal{L}_{feature}$ $\qquad\qquad\qquad$ ▷ detailed in Eq. (29)
14 $\quad\quad\quad$ $D' \leftarrow D' + \mathcal{L}_{feature} + \alpha\mathcal{L}_{structure} + \beta||A'||_2$
15 $\quad\quad$ **if** $t\%(\tau_1 + \tau_2) < \tau_1$ **then**
16 $\quad\quad\quad$ Update $\mathbf{X}' \leftarrow \mathbf{X}' - \eta_1\nabla_{\mathbf{X}'}D'$
17 $\quad\quad$ **else**
18 $\quad\quad\quad$ Update $\Phi \leftarrow \Phi - \eta_2\nabla_\Phi D'$
19 $\quad\quad$ Update $\boldsymbol{\theta}_{t+1} \leftarrow \text{opt}_{\boldsymbol{\theta}}(\boldsymbol{\theta}_t, \mathcal{S}, \tau_{\boldsymbol{\theta}})$ $\qquad$ ▷ $\tau_{\boldsymbol{\theta}}$ is the number of steps for updating $\boldsymbol{\theta}$
20 $\mathbf{A}' = \text{GEN}(\mathcal{Z}(N') \oplus \mathbf{X}' \oplus \mathbf{Y}'; \Phi)$
21 $\mathbf{A}'_{ij} = \mathbf{A}'_{ij}$ if $\mathbf{A}'_{ij} > 0.5$, otherwise 0
22 **Return:** $(\mathbf{A}', \mathbf{X}', \mathbf{Y}')$

---

### C.5   More explorations of the sensitivity of $\beta$

In Fig. 4(d), we demonstrate that SGDD is not sensitive to the coefficient $\beta$ on the YelpChi dataset. Here, we provide more experiments on the other datasets. In Fig. 6, we can see that increasing $\beta$ leads to the more sparsity of the condensed graph, and the corresponding performance does not have severe drop. These observations demonstrate the effectiveness of the regularity term in our objective.

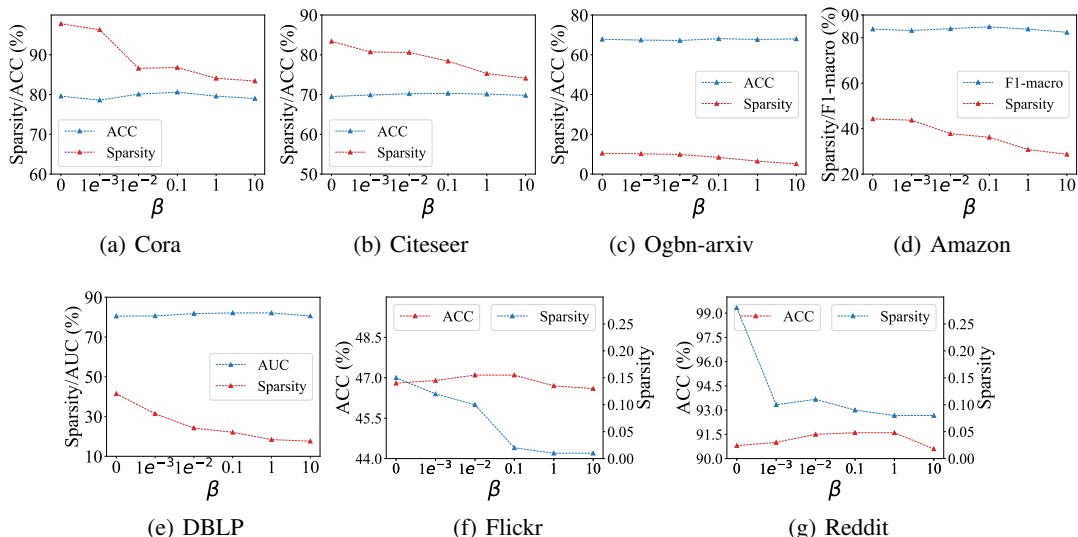

Figure 6: Evaluations of $\beta$ in seven datasets. For the Flickr and Reddit datasets, due to the datasets' low sparsity, we use the shared x-axis form for the Flickr and Reddit figures.

Table 7: Comparison of the cross-architecture generalization performance between GCond and SGDD on YelpChi. **Bold entries** are the best results. ↑/↓: our method show increase or decrease performance.

| C/T | APPNP GCond / SGDD | Cheby GCond / SGDD | GCN GCond / SGDD | SAGE GCond / SGDD | SGC GCond / SGDD |
|---|---|---|---|---|---|
| APPNP | 48.1 / **55.1**↑ | 46.5 / **57.4**↑ | 50.1 / **58.6**↑ | 46.7 / **57.1**↑ | 49.6 / **57.6**↑ |
| Cheby | 48.0 / **56.2**↑ | 45.9 / **56.8**↑ | 49.8 / **58.7**↑ | 46.8 / **58.3**↑ | 49.8 / **58.4**↑ |
| GCN | 47.6 / **56.5**↑ | 46.6 / **56.8**↑ | 48.6 / **59.7**↑ | 47.4 / **57.6**↑ | 50.1 / **57.4**↑ |
| SAGE | 46.7 / **57.6**↑ | 46.8 / **57.5**↑ | 48.9 / **58.7**↑ | 48.6 / **58.6**↑ | 48.9 / **58.6**↑ |
| SGC | 47.6 / **57.6**↑ | 47.7 / **57.2**↑ | 48.6 / **57.8**↑ | 47.4 / **59.0**↑ | 48.7 / **57.6**↑ |

## C.6 More cross-architecture experiments

In this subsection, we further report the results of cross-architecture experiments on the YelpChi. As shown in Tab. 7, our SGDD improves the performance in all cases, the average improvements compared to the GCond [37] is 9.6%, which indicates the strong generalization performance of the condensed graph by SGDD.

## C.7 Ablation of the sampling operation in OT distance

To further reduce condensing time and memory usage, we follow GCond [37] to sample from original graph **A** in the condensing stage (line 11 in the Algorithm 1). We evaluate the SGDD on various sample sizes, *i.e.*, {100, 500, 1000, 2000, 5000}, and report the corresponding performance and $SC$. As shown in Fig. 7, with the increase in the number of nodes, the performance obtains marginal improvements while the $SC$ is decreasing. However, larger sample sizes are not always beneficial to performance. In this study, the highest results are obtained when the sample size is 2000. This result empirically demonstrates the scalability of the structure learning module. Specifically, the module enables efficient condensing on large graphs.

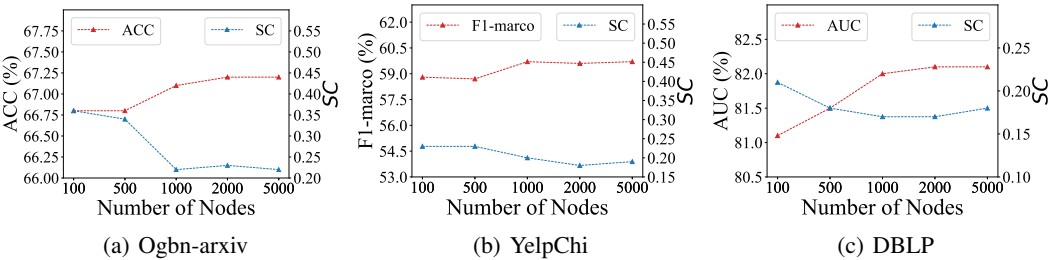

| (a) Ogbn-arxiv | (b) YelpChi | (c) DBLP |
|---|---|---|

Figure 7: Evaluation of the number of sampled nodes to the performance.

## C.8 More discussion of the condensed graphs.

We next show the comparison of condensed graphs and original graphs. As shown in Tab. 8, the condensed graphs obviously have fewer nodes and edges, while maintaining comparable accuracy, which shows the effectiveness of our SGDD . We also represent the visualizations of some condensed graphs. In Fig. 8, the black lines denote that edge weights are larger than 0.5 and the gray lines represent as smaller than 0.5. We can observe that there are some patterns in the condensed graph, *e.g.*, the homophily patterns on Cora and Citeseer. However, for the remaining datasets, the visualizations are inadequate in revealing the properties of the condensed graph, which proves the superiority of analyzing structure in spectral view.

## C.9 More discussion of the large graphs.

We conduct experiments on the Ogbn-mag datasets and show the details as follows: (note we transform the original heterogeneous graph to the homogeneous by ignoring difference of

Table 8: The comparison between condensed graphs and original graphs.

| | Citeseer, r=0.9% | | Cora, r=1.3% | | Ogbn-arxiv, r=0.5% | | Flickr, r=0.1% | | Reddit, r=0.1% | |
| --- | --- | --- | --- | --- | --- | --- | --- | --- | --- | --- |
| | Whole | SGDD | Whole | SGDD | Whole | SGDD | Whole | SGDD | Whole | SGDD |
| Accuracy | 70.7 | 69.5 | 81.5 | 79.6 | 71.4 | 65.3 | 47.1 | 47.1 | 94.1 | 90.5 |
| #Nodes | 3k3 | 60 | 2k7 | 70 | 169k | 454 | 44k | 44 | 153k | 153 |
| #Edges | 4k7 | 1k | 5k4 | 2k | 1,166k | 8k6 | 218k | 331 | 10,753k | 3k |
| Storage(MB) | 47.1 | 0.8 | 14.9 | 0.4 | 100.4 | 1.0 | 86.8 | 0.1 | 435.5 | 0.7 |

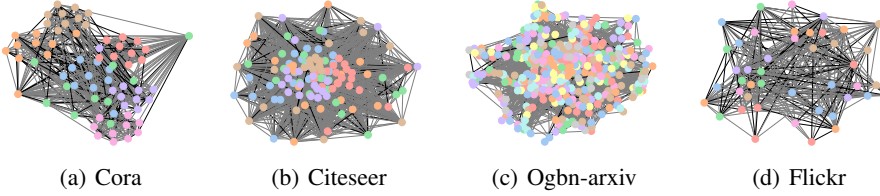

(a) Cora          (b) Citeseer          (c) Ogbn-arxiv          (d) Flickr

Figure 8: Visualizations of condensed graphs, the different colors on the graphs denote the classes.

