# OpenReview forum: "Does Graph Distillation See Like Vision Dataset Counterpart?"
_NeurIPS.cc/2023/Conference — NeurIPS 2023 poster_

### Official Review · Reviewer_FmUF · 2023-07-02

**Soundness:** 2 fair
**Presentation:** 3 good
**Contribution:** 2 fair
**Rating:** 4
**Confidence:** 3

**Summary:**

This paper mainly focuses on the Laplacian Energy Distribution (LED) shift problem of graph dataset condensation.

**Strengths:**

The pipeline figure is clear and straightforward.

The studied problem, how to condense a graph dataset, is relatively important.


**Weaknesses:**

Compared to existing studies that widely investigated the gradient matching, it seems that the major contribution of this paper is considering the LED shift. Why LED and how can LED benefit? The authors need to provide more analyses and explanation about their motivation.

In the Introduction, the authors mention “We empirically find a positive correlation between LED shift and the performance in cross-architecture settings”. This can only be considered as a finding. The motivation for introducing LED is ambiguous, especially given the correlation does not mean causality.

The authors simply compare their method  with Gcond and consider it as the SOTA. However, there are many works that have been proposed since Gcond. The authors should consider compare with them as well for a comprehensive evaluation of their method. For example, [1, 2, 3, 4, 5]

[1] Jin, Wei, et al. "Condensing graphs via one-step gradient matching." Proceedings of the 28th ACM SIGKDD Conference on Knowledge Discovery and Data Mining. 2022.

[2] Yu, Ruonan, Songhua Liu, and Xinchao Wang. "Dataset distillation: A comprehensive review." arXiv preprint arXiv:2301.07014 (2023).

[3] Liu, Chuang, et al. "Comprehensive graph gradual pruning for sparse training in graph neural networks." IEEE Transactions on Neural Networks and Learning Systems (2023).

[4] Zheng, Xin, et al. "Structure-free Graph Condensation: From Large-scale Graphs to Condensed Graph-free Data." arXiv preprint arXiv:2306.02664 (2023).

[5] Yang, Shuo, et al. "Dataset pruning: Reducing training data by examining generalization influence." arXiv preprint arXiv:2205.09329 (2022).


**Questions:**

see above.

**Limitations:**

see above.

---

> ### Author Rebuttal · Authors · 2023-08-09
>
> We sincerely thank the reviewer for the instructive questions. We make responses to the reviewer’s comments as follows.
>
> ## Q1: More analyses and explanations of motivation?
>
> A1:  Thanks for the question, we show the storyline of our paper as follows.
>
> - The core question in our paper is: “Does graph distillation see like vision dataset counterpart?“. Previous works follow the vision dataset distillation methods, which may lead to the following limitations: 1) the original graph structure information is not well preserved after condensing (see Fig. 1(b)); 2) In vision dataset distillation, gradient matching may entangle the synthetic dataset and architecture[86], which may be compounded on graph data (lines 43-54).
> - Then how to broadcast the original structure information into the condensed graph? In graph spectral theory [63, 1, 10, 62], the LED shift serves as a metric for measuring the differences between two graph structures. However, due to the significant size difference between the original and condensed graphs, directly calculating the LED shift is impractical. Inspired by the [63, 1], we analyze the LED shift between two graphs in the spectral view. We introduce the LED Shift Coefficient ($SC$) as a measure of shifts and empirically validate its consistency with performance. As optimizing $SC$ incurs substantial costs (see our response to Q4 of reviewer 5ivm), we further employ the Optimal Transport distance to approximate $SC$ during optimization. A detailed analysis is provided in lines 172 to 181.
> - Our SGDD approach achieves state-of-the-art results on 9 datasets. For example, in the YelpChi, we maintain 98.6% test accuracy while reducing the graph scale by 1,000 times.
> - Compared to previous methods, $SC$ is significantly reduced by SGDD. For example,  29.4% reduction on Reddit, 50.0% on YelpChi, 60.8% on Reddit, and 62.5% on Amazon (see Fig. 1(b, e), Fig. 5(b, e), and the response to Q4 of reviewer 5ivm).
>
> In summary:
>
> 1. The capability of the LED to convert the graph structure into a frequency domain distribution is essential for calculating the structure distance, particularly in cases where there is a substantial size difference.
> 2. Following the discussion on [63, 1, 10, 62], the LED is closely associated with the generalization performance of GNNs, providing valuable insights into the role of graph structure in the condensation process.
>
> **The correlation does not mean causality:** Thanks for the question. The correlation inspires us to design the SGDD method, which effectively broadcast the structural information from the original graph dataset to a condensed version. Our SGDD achieves SOTA in most cases, including new SOTA results on YelpChi and Amazon with improvements of 9.6% and 7.7%, respectively.
>
> Empirical results support the consistency between LED shift and generalization performance (see Fig. 1(b, e), Fig. 5(b, e), and response to Q4 of reviewer 5ivm). Compared to previous methods, our SGDD significantly improves performance while reducing $SC$.
>
> ---
>
> ## Q2: Considering compare with more baselines?
>
> A2: Thanks for the advice. We conduct experiments on 5 additional baselines.
>
> We first clarify that (1) the **public date (5 Jun) of SFGC [4] is behind the deadline for NeurIPS submission (17 May).** Nevertheless,  we would like to compare them in the revision. (2) the literature [2] is a survey on the computer vision area, we choose the representative method MTT [A] to compare with.
>
> - Note and details:
>     - We calculate the average performance of each method and use the $\Delta$ to indicate the relative difference of the SGDD to the other method.
>     - Since CGP [3] has no public code, we reproduce it in the evaluation.
>     - The Dataset Pruning [5] requires a solver in linear programming, we use the CPLEX [B] as the backend optimizer.
>
> We highlight the best-performing entries in **bold** and indicate the $\underline{\text{runner-ups}}$ with underlined values.
>
> |  | Citeseer($r$=1.8%) | Cora($r$=2.6%) | Ogbn-arxiv($r$=0.25%) | Flickr($r$=0.5%) | Reddit($r$=0.1%) | YelpChi($r$=0.1%) | Amazon($r$=0.2%) | Avg.  / $\Delta$ (%) |
> | --- | --- | --- | --- | --- | --- | --- | --- | --- |
> | Whole Dataset | 71.7±0.1 | 81.2±0.2 | 71.4±0.1 | 47.2±0.1 | 93.9±0.0 | 61.1±1.8 | 89.5±0.9 | 73.7 / - |
> | MTT [2, A] | 68.4±0.8 | 78.3±0.8 | 59.9±0.6 | 44.8±0.8 | 86.7±0.1 | 44.1±0.8 | 76.7±1.1 | 65.5 / **-5.8** |
> | CGP [3] | 67.4±1.3 | 77.6±0.8 | 61.1±0.7 | 45.4±0.4 | 86.6±1.8 | 49.6±0.2 | 77.4±1.8 | 66.4 / **-4.9** |
> | Dataset Pruning [5] | 66.8±1.8 | 74.6±1.9 | More than 3 days | More than 3 days | More than 3 days | 46.8±1.1 | 68.6±0.4 | 64.2 / - |
> | DosCond [1] | 69.8±0.3 | 79.4±0.7 | 58.8±1.1 | 46.3±0.4 | 88.6±0.5 | 47.6±0.3 | 77.4±0.8 | 66.8 / **-4.5** |
> | SFGC [4] | **72.4±0.4** | **81.7±0.5** | $\underline{66.1±0.4}$ | $\underline{47.0±0.1}$ | $\underline{90.0±0.3}$ | $\underline{44.7±0.1}$ | $\underline{77.5±0.7}$ | 68.4 / **-2.9** |
> | SGDD | $\underline{70.3±0.8}$ | $\underline{80.6±0.8}$ | **67.2±2.8** | **47.1±0.3** | **91.8±1.9** | **58.1±2.3** | **84.8±1.7** | 71.4 |
>
> Conclusions:
>
> - **Compared to Dataset Condensation methods [2, 1, 4], our method demonstrates an average improvement across all datasets. Notably, SGDD exhibits higher improvements on large datasets, highlighting the importance of broadcasting the structural information in condensing larger datasets.**
> - **Compared to dataset pruning methods [3, 5], the linear programming-based approach [5] fails in efficiency, and the mask-based method [3] may hinder condensed graph connectivity, resulting in poor performance.**
>
> Due to the limited time, we primarily focus on the node classification and anomaly detection tasks, we would like to show more experiments if the reviewer is interested.
>
> ---
>
> [A] George Cazenavett. et al. "Dataset Distillation by Matching Training Trajectories." CVPR  2022.
>
> [B] Cplex, I. I. User’s Manual for CPLEX. *International Business Machines Corporation*, 2009.

---

> > ### Author Response · Authors · 2023-08-17
> > **Further Discussions with Reviewer FmUF**
> >
> > Dear reviewer FmUF:
> >
> > Thanks for taking the time to review our paper. We detailed our motivation in the response and provide a more comprehensive study with additional 5 works. Hope our rebuttal has addressed your concerns.
> >
> > As the discussion period is nearing its end, please feel free to let us know if you have any other concerns. Thanks!

---

> > > ### Author Response · Authors · 2023-08-20
> > > **Looking forward to your reply!**
> > >
> > > Dear Reviewer FmUF,
> > >
> > > As we draw closer to the rebuttal deadline, I would like to inquire if you have any additional questions or concerns about our work. We greatly value your feedback. Thank you!
> > >
> > > Best,
> > >
> > > Authors from submission 1279

---

> > > > ### Author Response · Authors · 2023-08-21
> > > >
> > > > Dear Reviewer FmUF,
> > > >
> > > > Considering the limited time available, and in order to save the reviewer's time, we have concisely summarized our responses here.
> > > >
> > > > 1. ****More analyses and explanations of motivation:****
> > > >
> > > >     Question: The authors need to provide more analyses and explanations about their motivation.
> > > >
> > > >     Response: **The core question in our paper is: “Does graph distillation see like vision dataset counterpart?“.** We start from the point of researching on “structure”, and choose the spectral perspective. The step-by-step stroyline is listed in our rebuttal.
> > > >
> > > >     Question: The motivation for introducing LED is ambiguous, especially given the correlation does not mean causality.
> > > >
> > > >     Response: **The correlation inspires us to design the SGDD method,** which effectively broadcast the structural information from the original graph dataset to a condensed version. Our SGDD achieves SOTA in most cases, including new SOTA results on YelpChi and Amazon with improvements of 9.6% and 7.7%, respectively.
> > > >
> > > >     **Empirical results support the consistency between LED shift and generalization performance (see Fig. 1(b, e), Fig. 5(b, e), and response to Q4 of reviewer 5ivm)**. Compared to previous methods, our SGDD significantly improves performance while reducing $SC$.
> > > >
> > > > 2. ****Consider comparing with more baselines****
> > > >
> > > >     Question: The authors should consider compare with them as well for a comprehensive evaluation of their method. For example, [1, 2, 3, 4, 5].
> > > >
> > > >     Response: **We compared with 5 new baselines on 7 datasets,** including the dataset pruning methods and the latest graph condensation work SFGC (released later than NeurIPS submission deadline). **The results further proved the effectiveness of the graph dataset distillation as well as our method SGDD. We would like to add the experiment to our revisions.**
> > > >
> > > >
> > > > Conclusion(s):
> > > >
> > > > - *We are endeavored to answer the question “Does graph distillation see like vision dataset counterpart?” through the spectral perspective.*
> > > > - *Compared to other dataset pruning methods[2, A, 3, 5] and the latest graph condensation methods[1, 4], **our proposed SGDD still outperform them at least 2.9% in average, reflecting the importance of broadcasting the structural information in condensing datasets.***
> > > >
> > > > Since the discussion period is about to close in several hours, may I know if our responses addressed your concerns? Please feel free to let us know if you have any other concerns. Thanks!
> > > >
> > > > Best,
> > > >
> > > > Authors from submission 1279

---

### Official Review · Reviewer_UGAQ · 2023-07-06

**Soundness:** 3 good
**Presentation:** 3 good
**Contribution:** 3 good
**Rating:** 7
**Confidence:** 4

**Summary:**

This paper proposes a novel method called SGDD for condensing large-scale graph datasets while preserving the original structure information. The proposed method uses a graphon approximation method to broadcast the original structure as supervision for generating the condensed graph structure and optimizes it using an optimal transport method. The proposed SGDD achieves state-of-the-art results on various datasets and tasks.

**Strengths:**

(1) This paper uncovers the issue of Laplacian energy distribution shift during the condensation of graph datasets and shows that previous graph condensation methods that overlook the original structure information can lead to poor performance in cross-architecture generalization and specific tasks. This paper introduces a metric SC to quantify the Laplacian energy distribution issue.
(2) A new condensing paradigm in graph condensation. This paper leverages the graphon theory to guarantee the structure information consistency between the original and generated graph, which firstly involves the generated learning fashion in graph condensation. Consequently, the generated structure may capture the native properties of the original structure in comparison to current methods, which benefits the downstream tasks.
(3) Extensive experimental results demonstrate the effectiveness of the proposed method. This work includes extensive experiments on three node-level tasks as well as multiple ablation studies. All of the experiments demonstrate superior results. The most appealing result is that this method performs well in a cross-architecture setting, significantly improving the effectiveness of graph condensation.
(4) Potential Impact. When the data and model go bigger and bigger, graph condensation may have more applications and research meanings. As the applications and challenges related to graph condensation remain unidentified, this paper may set the stage for future exploration of this problem and inspire the communities to further investigate and may benefit the areas like pruning, LLM with graphs, and NAS.


**Weaknesses:**

(1) The motivation for using JS divergence in the calculation of SC is unclear. There are serval ways of calculating the distance of two distributions.
(2) In Figure.2, the author uses the BWGNN without specifying the difference between BWGNN and other backbones.
(3) There is another related method called DosCond[1] which is mentioned in the introduction but not involved in the baselines. The reason why not choosing this method for comparison should be specified.
[1] Jin, Wei, et al. “Condensing Graphs via One-Step Gradient Matching.” Proceedings of the 28th ACM SIGKDD Conference on Knowledge Discovery and Data Mining, 2022.


**Questions:**

(1) Please detailed the motivation in weakness.1 and weakness.2.
(2) Consider adding the baseline mentioned in weakness (3) or specify the reason why do not compare with this method.
(3) There is a high variance in the results of SGDD in Table 1. Would changing the backbone potentially improve the overall performance as shown in Table 1?


**Limitations:**

(1) In the real world, graphs may contain multiple types of edges, resulting in heterogeneity. However, the current learning methods may fail to capture this heterogeneity, leading to information loss.
(2) Lack of discussion on different backbones.

---

> ### Author Rebuttal · Authors · 2023-08-09
>
> We sincerely thank the reviewer for the detailed comments and insightful questions. We make responses to the reviewer’s comments as follows.
>
> ## Q1: The motivation for using JS divergence in the calculation of $SC$ is unclear?
>
> A1: Thanks for the comment. We list three commonly used distances in the table below and compared their characteristics.
>
> | Distance Type | Characteristics |
> | - | - |
> | Wasserstein distance | Wasserstein distance is often used in the situation where the two distributions are non-overlapping |
> | Kullback-Leibler divergence (KL) | KL-divergence is asymmetrical and can’t satisfy the requirement in comparing |
> | Jensen-Shannon divergence (JS) | It offers an intuitive perspective on the similarity or dissimilarity of two distributions in terms of their shapes. |
>
> Here we list the results on two graphs $G_a$ and $G_b$ condensed from the four datasets ($G_o$). We report the different distance and the average accuracy of them.
>
> Note we use the $\uparrow$, $\downarrow$, and - to denote the improve, decrease, and no change respectively.
>
> |  | Cora, | $r$=2.6% | Citeseer, | $r$=1.8% | Flickr, | $r$=0.5% | Reddit, | $r$=0.05% |
> | - | - | - | - | - | - | - | - | - |
> |  | $G_a$ vs $G_o$ | $G_b$ vs $G_o$ | $G_a$ vs $G_o$ | $G_b$ vs $G_o$ | $G_a$ vs $G_o$ | $G_b$ vs $G_o$ | $G_a$ vs $G_o$ | $G_b$ vs $G_o$ |
> | KL | 0.15 | 0.15 - | 0.16 | 0.32$\uparrow$ | 0.18 | 0.19$\uparrow$ | 0.21 | 0.22$\uparrow$ |
> | Wasserstein distance | 0.24 | 0.21 $\downarrow$ | 0.14 | 0.18$\uparrow$ | 0.26 | 0.17$\downarrow$ | 0.32 | 0.18$\downarrow$ |
> | JS | 0.11 | 0.24 $\uparrow$ | 0.27 | 0.36$\uparrow$ | 0.16 | 0.19$\uparrow$ | 0.33 | 0.42$\uparrow$ |
> | Avg. Acc. (%) | 74.1 | 68.2 $\downarrow$ | 68.2 | 64.2$\downarrow$ | 45.6 | 33.1$\downarrow$ | 90.6 | 88.7$\downarrow$ |
>
> Conclusion(s):
>
> - **While KL divergence and Wasserstein distance could theoretically be used, we observe that their consistency was inferior to that of JS divergence in this specific case. As a result, we chose to utilize JS divergence in our experiments.**
>
> ## Q2: The difference between BWGNN and other backbones?
>
> In Figure.2, the author uses the BWGNN without specifying the difference between BWGNN and other backbones
>
> A2:  Thanks for the question.
>
> **Motivation:**  We select the BWGNN [63] in order to examine the impact of the frequency response of each backbone on the Laplacian energy distribution (LED) of the condensed graph, it is crucial to eliminate the influence of differences among the backbones themselves. Therefore, we choose the frequency-adaptive BWGNN.
>
> **Brief introduction to BWGNN:** the BWGNN mainly uses the Beta wavelet, which has the probability density function of Beta distribution admits:
>
> $$
> \beta_{p, q}(w)= \begin{cases}\frac{1}{B(p+1, q+1)} w^p(1-w)^q, & \text { if } w \in[0,1] \\ 0, & \text { otherwise }\end{cases}
> $$
>
> Where $p, q \in \mathbb{R}^{+}$ and $B(p+1, q+1) = p!q!/(p+q+1)!$ is a constant, the BWGNN adopt $B^*_{(p, q)} (w) = \frac{1}{2}B_{p,q} (\frac{w}{2})$ to cover the complete spectral range of $L$. Thus we can use a different parameter setting to change the frequency response of BWGNN (see 3.2 Beta Wavelet on Graph on [63]).
>
> **Difference to other backbones:** Following  [1], we list the difference between BWGNN and the other backbones as follows.
>
> | Backbones | Low-pass support | high-pass support | Controllable spectral range |
> | - | - | - | - |
> | SGC | $\checkmark$ | $\times$ | $\times$ |
> | GCN | $\checkmark$ | $\times$ | $\times$ |
> | ChebNet | $\checkmark$ | $\checkmark$ | $\times$ |
> | GAT | $\checkmark$ | $\checkmark$ | $\times$ |
> | BWGNN | $\checkmark$ | $\checkmark$ | $\checkmark$ |
>
> Conclusion(s):
>
> -  **Compared to other backbones, BWGNN not only provides support for all-pass filtering but also offers controllability, which is convenient for applying it to 9 datasets.**
>
> ## Q3: The experiment results about DosCond?
>
> A3: Thanks for the suggestion. We conduct experiments on node classification and anomaly detection tasks on DosCond [29] and show the results in the following table.
>
> - **Note:** We use $\Delta$ to indicate the performance improvement from method DosCond to SGDD.
>
> |  | Citeseer($r$=1.8%) | Cora($r$=2.6%) | Ogbn-arxiv($r$=0.25%) | Flickr($r$=0.5%) | Reddit($r$=0.1%) | YelpChi($r$=0.1%) | Amazon($r$=0.2%) |
> | - | - | - | - | - | - | - | - |
> | Whole Dataset | 71.7±0.1 | 81.2±0.2 | 71.4±0.1 | 47.2±0.1 | 93.9±0.0 | 61.1±1.8 | 89.5±0.9 |
> | DosCond [3] | 69.8±1.8 | 79.4±1.1 | 58.8±1.3 | 46.3±0.4 | 88.6±0.5 | 47.6±0.3 | 77.4±0.8 |
> | SGDD | 70.3±0.8 | 80.6±0.8 | 67.2±2.8 | 47.1±0.3 | 91.8±1.9 | 58.1±2.3 | 84.8±1.7 |
> | $\Delta$(%) | 0.5 $\uparrow$ | 1.2 $\uparrow$ | 8.4$\uparrow$ | 0.8$\uparrow$ | 3.2$\uparrow$ | 10.5$\uparrow$ | 7.3$\uparrow$ |
>
> Conclusion(s):
>
> - **Our method SGDD demonstrates superior performance compared to DosCond, particularly on the YelpChi and Amazon datasets.**
>
> ## Q4: Would changing the backbone potentially improve the overall performance?
>
> A4: Thanks for the question. We conduct the experiments on three datasets: Reddit, Ogbn-arxiv, and Flickr with 6 different backbones.
>
> - **Note:** We use the $\uparrow$ and $\downarrow$ to indicate the increase and the decrease to the default GCN respectively.
>
> | Acc. (%) | GCN | APPNP | Cheby | SAGE | SGC | GAT |
> | - | - | - | - | - | - | - |
> | Reddit, $r$=0.05% | 91.8 | 91.4/0.4$\downarrow$ | 92.1/0.3$\uparrow$ | 90.6/1.2$\downarrow$ | 91.9/0.1$\uparrow$ | 92.0/0.2$\uparrow$ |
> | Ogbn-arxiv, $r$=0.25% | 67.2 | 66.4/0.8$\downarrow$  | 66.8/0.4$\downarrow$ | 66.5/0.7$\downarrow$ | 66.4/0.8$\downarrow$ | 66.4/0.8$\downarrow$ |
> | Flickr, $r$=0.50% | 47.1 | 47.2/0.1$\uparrow$  | 46.2/0.9$\downarrow$ | 46.8/0.3$\downarrow$ | 47.0/0.1$\downarrow$ | 47.2/0.1$\uparrow$ |
>
> Conclusion(s):
>
> - **The range of upward and downward performance is not particularly large in our case, as it falls within a specific range between -0.8 and 0.3.**

---

> > ### Comment · Reviewer_UGAQ · 2023-08-18
> >
> > Thanks the authors for their clear rebuttals and efforts, which have addressed my concerns and problems. Especially the extensive experiments on various datasets validate their effectiveness and superiority compared to baselines.  In summary, this paper proposes a novel method called SGDD for condensing large-scale graph datasets while preserving the original structure information, and achieves state-of-the-art results. It is worth noting that data condensation has not been extensively explored in the field of graph learning, and this work addresses the problem from a novel perspective of graph structure, which will play a leading role in the field of work. I suggest this paper to be accepted.

---

> > > ### Author Response · Authors · 2023-08-19
> > >
> > > Dear Reviewer UGAQ,
> > > We would like to express our sincere gratitude to the reviewer UGAQ for endorsing our work and providing constructive suggestions.
> > > Yes, we find that graph dataset distillation is largely different from vision dataset distillation, so we design our method.
> > > Thanks again for the time and effort in reviewing our work.

---

### Official Review · Reviewer_9hTZ · 2023-07-07

**Soundness:** 3 good
**Presentation:** 3 good
**Contribution:** 3 good
**Rating:** 4
**Confidence:** 4

**Summary:**

The paper proposes a novel approach for graph dataset distillation called Structure-broadcasting Graph Dataset Distillation (SGDD). The authors explicitly consider the impact of the original structure information on graph condensation and demonstrate that their approach achieves state-of-the-art results on 9 datasets, showing superior performance in cross-architecture settings and specific tasks. Overall, the paper presents a significant contribution to the field of graph dataset distillation.

**Strengths:**

- The paper proposes a novel approach for graph dataset distillation that explicitly considers the impact of the original structure information on graph condensation. This is a significant departure from existing methods that primarily focus on optimizing the feature matrices of condensed graphs while overlooking the structure information.
- The authors introduce the concept of Substantial Laplacian Energy Distribution (LED) shifts and demonstrate that previous works suffer from such shifts, leading to poor performance in cross-architecture generalization and specific tasks. The authors propose SGDD as a solution to this problem, which is a novel and original contribution.
- The authors provide a thorough analysis of the proposed approach, including theoretical analysis and empirical evaluation on 9 datasets. The results demonstrate that SGDD consistently outperforms existing state-of-the-art methods, indicating the high quality of the proposed approach.


**Weaknesses:**

- [Actual Computation Savings] The authors distill the graph dataset. However, the authors only provide the distilled dataset's size ratio to original dataset size. It would be helpful to provide the actual computation savings in terms of time and memory usage.

- [The Cost during Graph Distillation]  The cost of graph distillation for condensed small graph dataset is recommended to be provided. Since the used benchmark graph datasets are less computationally expensive than datasets in computer vision and natural language processing, the cost of graph distillation period will reflect whether the dataset is necessary for practical use.

- [Comparison with Other Methods]  The paper provides a comparison of the proposed method with other methods in terms of performance and size ratio (e.g., Table 1). However, it would be beneficial to also compare the cost of graph distillation for the proposed method with other methods. This information would provide a more comprehensive understanding of the practicality of the proposed approach compared to other methods.

**Questions:**

Questions: Please see weaknesses. I would like to update my evaluation after the discussion.


**Limitations:**

N.A.

---

> ### Author Rebuttal · Authors · 2023-08-09
>
> We appreciate reviewer 9hTZ’s constructive feedback and are glad that the reviewer finds our work novel. We answer the questions one by one as follows. Hope it can address the reviewer’s concern.
>
> ### Q1: [Actual Computation Savings]
>
> A1: Thanks for the suggestion. We conduct the experiments on 5 datasets and show results as follows.
>
> **Storage Saving:** We present the basic statics of datasets and the storage saving in the following table.
>
> |  | Citeseer, |   $r$=0.9% | Cora,  |  $r$=1.3% | Ogbn-arxiv,  |  $r$=0.5% | Flickr,  | $r$=0.1% | Reddit,   | $r$=0.1% |
> | --- | --- | --- | --- | --- | --- | --- | --- | --- | --- | --- |
> |  | Whole | SGDD | Whole | SGDD | Whole | SGDD | Whole | SGDD | Whole | SGDD |
> | Accuracy (%) | 70.7 | 69.5 | 81.5 | 79.6 | 71.4 | 65.3 | 47.1 | 47.1 | 93.9 | 90.5 |
> | #Nodes | 3,327 | 60 | 2,708 | 70 | 169,343 | 454 | 44,625 | 44 | 232,965 | 153 |
> | #Edges | 4,732 | 1,434 | 5,429 | 2,131 | 1,166,243 | 8,681 | 218,140 | 331 | 57,307,946 | 3,427 |
> | Storage (MB) | 47.1 | 0.8 **$\downarrow$ 58.8X** | 14.9 | 0.4 **$\downarrow$ 37.2X** | 100.4  | 1.0 **$\downarrow$ 100.4X** | 86.8 | 0.1 **$\downarrow$ 868X** | 435.5 | 0.7 **$\downarrow$ 622X** |
>
> Conclusion(s):
>
> - **We achieved a storage saving of 868X in Flickr, 622X in Reddit, 100.4X in Ogbn-arxiv, 58.8X in Citeseer, and 37.2X in Cora, respectively.**
>
> **Computing Saving:** We calculate the computing saving in the following table. All results are obtained with the repetition of 5 runs (X denotes times).
>
> |  | Whole Training Time(min) | Condensed  Training Time(min) | Acceleration Rate | Whole  Training Memory (GB) | Condensed  Training Memory(GB) | Compression Rate |
> | --- | --- | --- | --- | --- | --- | --- |
> | Cora | 18.6 ± 2.1 | 0.4 ± 0.1 | **$\downarrow$46.5X** | 3.2 ± 0.8 | 0.8 ± 0.1 | **$\downarrow$4.0X** |
> | Citeseer | 32.7 ± 5.8 | 0.8 ± 0.2 | **$\downarrow$40.8X** | 3.1 ± 0.6 | 0.8 ± 0.1 | **$\downarrow$3.9X** |
> | Reddit | 56.8 ± 10.1 | 1.1 ± 0.1 | **$\downarrow$51.6X** | 22.3 ± 1.5 | 1.4 ± 0.1 | **$\downarrow$15.9X** |
> | Flickr | 23.4 ± 4.2 | 1.1 ± 0.3 | **$\downarrow$23.0X** | 8.4 ± 0.8 | 1.1 ± 0.2 | **$\downarrow$7.6X** |
> | Ogbn-arxiv | 48.7 ± 12.1 | 1.2 ± 0.3 | **$\downarrow$40.5X** | 17.4 ± 1.8 | 1.2 ± 0.1 | **$\downarrow$14.5X** |
>
> Conclusion(s):
>
> - **Our method achieves a speedup of at least 23.0X and a compression ratio of at least 3.9X.**
>
> ### Q2: [The Cost during Graph Distillation]
>
> A2: Thanks for the suggestion. We compared the costs of the graph distillation process and the costs of training on the whole graph. The results are shown in the following table.
>
> We conduct experiments on 5 datasets and report the average computing time and memory. All experiments are repeated 5 times.
>
> |  | Whole Training Time(min) | Distillation Time(min) | Time Cost Rate | Whole  Training Memory (GB) | Distillation  Memory(GB) | Computing Cost Rate |
> | --- | --- | --- | --- | --- | --- | --- |
> | Cora | 18.6 ± 2.1 | 46.1 ± 4.8 | **$\uparrow$2.4X** | 3.2 ± 0.8 | 3.8 ± 0.6 | **$\uparrow$1.1X** |
> | Citeseer | 32.7 ± 5.8 | 53.8 ± 3.6 | **$\uparrow$1.6X** | 3.1 ± 0.6 | 4.1 ± 0.7 | **$\uparrow$1.3X** |
> | Reddit | 56.8 ± 10.1 | 120.8 ± 20.6 | **$\uparrow$2.1X** | 22.3 ± 1.5 | 28.6 ± 1.6 | **$\uparrow$1.3X** |
> | Flickr | 23.4 ± 4.2 | 67.3 ± 10.7 | **$\uparrow$2.8X** | 8.4 ± 0.8 | 11.7 ± 0.4 | **$\uparrow$1.4X** |
> | Ogbn-arxiv | 48.7 ± 12.1 | 140.3 ± 3.6 | **$\uparrow$2.8X** | 17.4 ± 1.8 | 22.7 ± 0.7 | **$\uparrow$1.3X** |
>
> Conclusion(s):
>
> - **Despite the cost of our method being 1.6X-2.8X higher, considering the acceleration rate (23-51X) and compression rate (3.9-15.9X) when condensation is finished, it remains highly valuable.**
>
> ### Q3: [Comparison with Other Methods]
>
> A3: Thanks for the suggestion. We compare our method SGDD and the other 6 baselines on three datasets (Ogbn-arxiv, Reddit, and Flickr), the average results (5 runs) are shown in the following table.
>
> - **Note:** We use the **Coarsening*** method as the base method to compare and calculate the time cost rate as well as the performance comparison.
>
> |  | Random | Herding | K-Center | Coarsening* | GDC | GCond | SGDD | Whole |
> | --- | --- | --- | --- | --- | --- | --- | --- | --- |
> | Avg. Time (min) | - | 10.2 ± 1.2 | 24 ± 1.2 | 61.7 ± 4.2 | 184.3 ± 10.4 | 160.2 ± 13.8 | 109.4 ± 10.2 | 42.7 ± 5.2 |
> | **Time Cost Rate** | - | **0.1X** | **0.3X** | **1.0X** | **3.0X** | **2.6X** | **1.7X** | - |
> | Avg. Computing (GB) | - | - | - | 7.4± 0.8 | 14.3 ± 1.2 | 14.8 ± 1.1 | 16.0 ± 1.4 | 11.5 ± 2.1 |
> | **Computing Cost Rate** | - | - | - | **1.0X** | **1.9X** | **2.0X** | **2.1X** | - |
> | Avg. Performance | 48.1 ± 2.1 | 54.6 ± 1.3 | 51 ± 0.7 | 56.6 ± 1.4 | 65.1 ± 0.8 | 66.6 ± 1.2 | 68.6 ± 0.7 | 73.1 ± 1.8 |
> | **Performance Comparison (%)** | **-8.5** | **-2.0** | **-5.6** | **0.0** | **+8.5** | **+10.0** | **+12.0** | - |
>
> Conclusions:
>
> - **Compare with heuristic methods: Heuristic methods have lower costs but much worse performance (-8.5% to -2.0%), indicating their limitations in condensation tasks.**
> - **Compare with other condensation methods: SGDD only introduces a slightly higher computational cost than GDC and GCond (2.1X vs 1.9X and 2.0X), but achieves faster convergence (1.7X vs 2.6X and 3.0X), which can be attributed to the benefits of the better-learned structure. Meanwhile, SGDD achieves higher results faster than GDC and GCond.**
>
> We hope the above response could address the concerns and would like to show more experiments in the discussion period if the reviewer is interested.

---

> > ### Author Response · Authors · 2023-08-17
> > **Further Discussions with Reviewer 9hTZ**
> >
> > Dear Reviewer 9hTZ:
> >
> > Thank you so much again for your time and efforts in assessing our paper. Hope our additional experiments on actual saving have addressed your concerns. We are happy to discuss with you further if you have other concerns. Thanks for helping improve our paper!

---

> > > ### Comment · Reviewer_9hTZ · 2023-08-17
> > >
> > > Thank you for your careful answer! I have gone through it.

---

> > > > ### Author Response · Authors · 2023-08-19
> > > > **Looking forward to the updated evaluation**
> > > >
> > > > Dear Reviewer 9hTZ,
> > > >
> > > > Thanks for the response. Based on your reviews, we are looking forward to your updated evaluations of our work.

---

> > > > > ### Author Response · Authors · 2023-08-20
> > > > >
> > > > > Dear Reviewer 9hTZ,
> > > > >
> > > > > As we are approaching the rebuttal deadline in less than two days, may I know if you have other concerns regarding our work? We truly value your insights and looking forward to your feedback!
> > > > >
> > > > > Best,
> > > > >
> > > > > Authors from Submission 1279

---

> > > > > > ### Author Response · Authors · 2023-08-21
> > > > > >
> > > > > > Dear Reviewer 9hTZ,
> > > > > >
> > > > > > Considering the limited time available, and in order to save the reviewer's time, we summarized our responses here.
> > > > > >
> > > > > > 1. **[Actual Computation Savings]**
> > > > > >
> > > > > >     Question: It would be helpful to provide the actual computation savings in terms of time and memory usage.
> > > > > >
> > > > > >     Response: We conduct the experiments on 5 datasets, and the results show that **our method achieves a speedup of at least 23.0X and a compression ratio of at least 3.9X.**
> > > > > >
> > > > > > 2. **[The Cost during Graph Distillation]**
> > > > > >
> > > > > >     Question: The cost of graph distillation for condensed small graph dataset is recommended to be provided.
> > > > > >
> > > > > >     Response: We conduct experiments to show the actual cost of the distillation process. **The results show that our SGDD require 1.6X-2.8X time to the naive training when condensing.**
> > > > > >
> > > > > > 3. **[Comparison with Other Methods]**
> > > > > >
> > > > > >     Question: It would be beneficial to also compare the cost of graph distillation for the proposed method with other methods.
> > > > > >
> > > > > >     Response: We conduct experiments on 6 other baselines on three datasets. From the table in the rebuttal, **our method only introduces a slightly higher computational cost than GDC and GCond (2.1X vs 1.9X and 2.0X), but achieves faster convergence (1.7X vs 2.6X and 3.0X).**
> > > > > >
> > > > > >
> > > > > > Conclusion(s):
> > > > > >
> > > > > > - *While our method's cost is **1.6X-2.8X** higher, its value is evident when considering the acceleration rate of **23-51X** and compression rate of **3.9-15.9X** upon completion of the condensation process. For instance, training Ogbn-arxiv requires **17.4 GB** of memory at the beginning, but this can be reduced to just **1.2 GB** after condensation. **This makes our method highly beneficial for users with limited resources as well as boosting scenarios like neural architecture search and incremental learning.***
> > > > > >
> > > > > > The author-reviewer discussion period will be closed less than hours, may I know if there are any other concerns? We truly value your insightful comment!

---

### Official Review · Reviewer_ZZ8V · 2023-07-11

**Soundness:** 3 good
**Presentation:** 4 excellent
**Contribution:** 3 good
**Rating:** 6
**Confidence:** 2

**Summary:**

The paper investigates the effects of structural information in graph condensation methods. The authors claim that by maintaining the original structure during condensation using a newly formulated method called Structure-broadcasting Graph Dataset Distillation (SGDD), they are able to achieve more refined results on 9 data sets thereby significantly reducing Laplacian Energy Distribution (LED) shift.

**Strengths:**

This paper addresses an important question regarding the impact of structural information. The authors conduct a comprehensive analysis from the spectral domain and empirically identify significant shifts in Laplacian Energy Distribution (LED), which ultimately result in poor performance in cross-architecture generalization and specific tasks.

The proposed method demonstrates effectiveness in the experiments, and empirical analysis confirms the efficacy and necessity of the proposed designs.

Overall, the presentation of the paper is also commendable

**Weaknesses:**

large datasets in the Open Graph Benchmark (OGB) is more effective in demonstrating the efficacy of these methods.

**Questions:**

I don't have question regarding this paper.

**Limitations:**

Need experiment for large datasets

---

> ### Author Rebuttal · Authors · 2023-08-09
>
> We sincerely thank the reviewer for the insightful comment. We make responses as follows.
>
> **Q1:** Need experiment for large datasets.
>
> A1: Thanks for the reviewer’s suggestion.
>
> **Dataset Statics:**
>
> - As illustrated in the table below, there are four datasets that are significantly larger than Ogbn-arxiv.
>
> | Datasets | #Nodes | #Edges | #Classes | Metric |
> | --- | --- | --- | --- | --- |
> | Ogbn-arxiv | 169,343 | 1,166,243 | 40 | Accuracy |
> | Ogbn-mag | 1,939,743 | 21,111,007 | 349 | Accuracy |
> | Ogbn-proteins | 132,534 | 3,561,252 | 112 (Multi-label) | ROC-AUC |
> | Ogbn-products | 2,449,029 | 61,859,140 | 47 | Accuracy |
> | Ogbn-papers100M | 111,059,956 | 1,615,685,872 | 172 | Accuracy |
>
> We conduct experiments on the Ogbn-mag datasets and show the details as follows.
>
> **Results on Ogbn-mag:**
>
> Note: we report the results on SGDD and the other 3 baselines on 7 different condensation ratios ($r$). The accuracy on the whole dataset is 30.4 [A].
>
> | Condensation Ratio($r$) | Random | K-Center | GCond | SGDD |
> | --- | --- | --- | --- | --- |
> |  0.0001% | 0.9 | 5.7 | 15.3 | 18.1 |
> | 0.0002% | 1.1 | 6.8 | 15.4 | 18.3 |
> | 0.0003% | 1.1 | 6.9 | 15.4 | 18.4 |
> | 0.0004% | 1.4 | 7.1 | 15.3 | 18.7 |
> | 0.0005% | 1.5 | 6.4 | 15.4 | 18.7 |
> | 0.001% | 1.4 | 8.7 | 15.4 | 18.8 |
> | 0.002% | 1.5 | 10.4 | 15.1 | 18.8 |
>
> Conclusion(s):
>
> - **On the large-scale dataset Ogbn-mag, our method achieves non-trivial improvements compared to other baselines.**
>
> Due to computational resource limitations, we are currently running experiments on additional datasets. We would like to provide it in the discussion period if the reviewer is interested.
>
> ---
>
> [A] Hu et al. Open Graph Benchmark: Datasets for Machine Learning on Graphs. NeurIPS 2021.

---

> > ### Author Response · Authors · 2023-08-19
> > **Further experiments on Ogbn-products**
> >
> > Dear reviewer ZZ8V:
> >
> > Thanks for your patience, we finished our experiments on Ogbn-products, and show the table as follows. We will add these results in the revision.
> >
> > **Results on Ogbn-products:**
> >
> > Note: we report the results of SGDD and the other 3 baselines on 7 different condensation ratios($r$). We use the $\Delta$ to denote the improvements for our proposed SGDD to K-Center.
> >
> > | Condensation Ratio ($r$) | Random | K-Center | GCond [B] | SGDD | $\Delta$ (%) |
> > | --- | --- | --- | --- | --- | --- |
> > | 0.0001% | 18.8 | 32.6 | 36.5 | 36.8  | $\uparrow$4.1 |
> > | 0.0002% | 18.4 | 34.7 | 36.4 | 36.8 | $\uparrow$2.0 |
> > | 0.0003% | 21.5 | 35.8 | 36.4 | 38.8 | $\uparrow$3.0 |
> > | 0.0004% | 23.8 | 35.6 | 36.9 | 38.6 | $\uparrow$3.0 |
> > | 0.0005% | 25.4 | 35.7 | 37.6 | 38.8 | $\uparrow$3.1 |
> > | 0.001% | 34.8 | 35.4 | 38.2 | 40.1 | $\uparrow$4.7 |
> > | 0.002% | 35.2 | 36.2 | 38.8 | 40.3 | $\uparrow$4.0 |
> >
> > Conclusion(s):
> >
> > - **Our proposed SGDD consistently outperforms other baselines across all condensation ratios with no meticulous tuning. Notably, it improves accuracy by 4.7% over K-Center at the 0.001% ratio, and by 1.8% and 5.3% over GCond [B] and Random, respectively.**
> >
> > We eagerly await your feedback to know if our experiments have adequately addressed your concerns. Please feel free if you have other questions. Thanks for your constructive suggestion again!
> >
> > [B] Wei Jin, et.al. Graph condensation for graph neural networks. ICLR 2022.

---

> > ### Comment · Reviewer_ZZ8V · 2023-08-20
> >
> > I would like to thank the authors for the detailed response. It helps clarify the work. Regrading the scores, it remains valid to reflect the quality of the work.

---

> > > ### Author Response · Authors · 2023-08-20
> > >
> > > Dear Reviewer ZZ8V, we are truly grateful for your acknowledgment regarding our additional evaluation. We would like to express our sincere thanks once again for the reviewer’s work on reviewing our work!

---

### Official Review · Reviewer_5ivm · 2023-07-12

**Soundness:** 2 fair
**Presentation:** 2 fair
**Contribution:** 2 fair
**Rating:** 5
**Confidence:** 3

**Summary:**

This paper proposes a new graph condensation framework called SGDD. The authors argue that existing methods overlook the structure information of the original graph during the condensation. And thus they propose to 1) use the Laplacian Energy Distribution (LED) shift to indicate the generalization performance of the condensed graph, and 2) minimize the LED shift (they claim it’s equivalent to the OT distance between the original and the condensed graphs) to preserve the structure information. Experiments on different datasets over node classification, anomaly detection and link prediction tasks show the effectiveness of the proposed method.

**Strengths:**

- The motivation of preserving the structure information is clear and reasonable.
- The experiments seem relatively sufficient and the results are good.

**Weaknesses:**

My major concern is the soundness of this paper:
- Some claims seem to be problematic. For example, in line 176-179, the authors claim that ‘minimizing the LED shift is equivalent to minimizing the distance of Laplacian pseudo-inverse matrices [65]’ and ‘Following the previous work [78], minimizing such distance can be further approximated to optimizing a free parameter $P$ in Eq(6)’. However, the equivalence in [65] only applies to the objective using a specific GW distance and the approximation in [78] only applies to another OT distance based objective. So the two claims cannot hold simultaneously. What’s worse, the authors do not describe which OT distance is used in their original objective function. Then how could we conclude that ‘minimizing the upper bound of Eq. (7) is equal to optimizing the $L_{structure}$ on Eq. (5)’ in line 189?
- The writing is not very easy to follow. The mixed use of ‘LED shift’, ‘LED shift coefficient’, ‘SC’ is very confusing (e.g., in Def2 and the caption of Fig2). And it’s not clear what’s the role of the proposed LED shift coefficient. This coefficient is neither used to derive the objective function, nor adopted as a performance metric in the tables for comparison.

=============================
Post-rebuttal: The author responses have basically solved my concerns about the soundness.

**Questions:**

Please see the above part.

**Limitations:**

None.

---

> ### Author Rebuttal · Authors · 2023-08-09
>
> We sincerely thank the reviewer for the detailed comments and insightful questions. We make responses as follows.
>
> ## **Q1:** The conflict of assumptions from the references [65] and [78].
>
> **A1:** Thanks for the comment. There are primarily two types of graph optimal transport distances: Gromov-Wasserstein (GW) and Graph Optimal Transport (GOT). We present the differences between them as follows.
>
> | Optimal Transports | Core Technologies | Famous Methods |
> | - | - | - |
> | GW | (1) GW is defined using a customized distance function that evaluates pairs of vertices in a graph [65]. (2) It quantifies the major alterations to the graph that significantly impact the distance function [A, B]. |FGW [65]、S-GWL[A]、SGW [B]|
> |GOT | (1) GOT is defined based on the graph signal [A]. (2) It primarily captures changes to the graph that significantly affect the eigenvectors of the Laplacian with small eigenvalues [78]. |FGOT [C]、COPT [78]、GOT [D]|
>
> FGW [65] belongs to the GW methods, while COPT [78] is a GOT-related method. Our proposed SGDD also relies on the graph signal $X$ of vertices and utilizes the Laplacian pseudo-inverse operation like [D], so our SGDD is a kind of GOT-related method. **After we checked our source code of Latex, there was a mistake in the reference GW-based method [65]. We update Line 175 to 177 as follows:**
>
> ***Optimal transport distance.** To address the above issue, we utilize optimal transport (OT) [78, D] to efficiently optimize the shift of LEDs. Specifically, minimizing the LED shift is equivalent to minimizing the distance of Laplacian pseudo-inverse matrices[D].*
>
> **We will carefully check the content related to this issue in revision.**
>
> ---
>
> [A] Hongteng Xu et. al. Scalable Gromov-Wasserstein Learning for Graph Partitioning and Matching. NeurIPS 2019.
>
> [B] Titouan et. al. Sliced Gromove-Wasserstein. NeurIPS 2019.
>
> [C] Maretic et. al. FGOT: Graph Distances based on Filters and Optimal Transport. AAAI 2021.
>
> [D] Petric Maretic et. al. GOT: an optimal transport framework for graph comparison. NeurIPS 2019.
>
> ## **Q2**: Why minimizing Eq. (7) is equal to optimizing the $L_{structure}$ on Eq. (5)?
>
> A2: Thanks for the question. First, considering the typically large different shapes between the original $\mathrm{A}$ and the condensed $\mathrm{A’}(N' \ll N)$ (line 160), the $A’$ here is the “graphon” $W_{\mathbf{A}}'$ with $N'$ nodes as an approximation to the oracle graphon $W_A$ [74], i.e., the abstract of the original graph. Thus Eq. (5) can be written as:
>
> $$ \mathcal{L}_{\textbf{structure}} = \mathrm{Distance} (W_A', W_A).$$
>
> In our settings, we replace the original graph $\mathrm{A}$ with $W_{\mathrm{A}}$ as $\mathrm{A}$ is an actual observed value of $W_A$ with $N$ nodes[74, 75]. Second, in Appendix B.2, the upper bound of Eq. (7) is the cut distance of two graphons:
>
> $$ \delta_{\square}(W_{\mathrm{A}}', {W_\mathrm{A}}). $$
>
> **As the cut distance $\delta_{\square}$ here can be regarded as a specific $\mathrm{Distance}$, the upper bound in Eq. (7) and** $L_{structure}$ in **Eq.(5) can be recognized equally.**
>
> We will make it clear and check similar issues in the revision. Thanks again for the question.
>
> ## **Q3:** Mixed use of the concepts of LED shift, LED shift coefficient, and $SC$.
>
> A3: Thanks for the comment. Assume that we have two graphs, we introduce the definitions of LED, LED shift, LED shift coefficient, and SC as follows,
>
> **LED** is the distribution of the post-graph Fourier-transformed graph signal of the graph.
>
> **LED shift** is the phenomenon that denotes the differences between these two graphs’ LED.
>
> **LED shift coefficient** reflects the divergence between these two graphs. We provide the detailed formulation in Eq. 4 (lines 144 to 147).
>
> $SC$ represents an abbreviation of the LED Shift Coefficient. We claim it in line 143 and  Eq. 4.
>
> We are sorry about the confusion and will make it clear in the revision.
>
> ## **Q4**: Where the $SC$ be used.
>
> A4: Thanks for the question.
>
> 1. As mentioned in lines 172 to 174, $SC$ requires the eigenvalue decomposition process, which is extremely time-consuming $O(N^3)$, particularly when dealing with large graphs. We show the estimated time cost per run of $SC$ as follows.
>
> |  | Cora, $r$=1.3% | Citeseer, $r$=0.9% | Reddit, $r$=0.1% | Flickr, $r$=0.1% | Ogbn-arxiv, $r$=0.5% |
> | - | - | - | - | - | - |
> | $SC$ (h) | 18.0 | 21.0 | 130.0 | 65.0 | 78.0 |
> | OT (h) | 0.8 | 0.9 | 2.1 | 1.1 | 1.8 |
> 1. Since we aim at alleviating the LED shift, the GOT-related OT distance serves as an approximate indicator ($O(N^2K)$($K \le N^{0.373}$)  [78]), which is more practical. As shown in the above table, OT is 61.9X faster than $SC$ on Reddit, 59.0X faster on Flickr, 43.3X faster on Ogbn-arxiv, 22.5X faster on Cora, and 23.3X faster on Citeseer, respectively.
> 2. By leveraging the OT distance, we achieve consistent reduction on $SC$ (see Figure 1(b, e) and Figure 5(b, e)). We provide more results as follows.
>
> Note: we use the $\downarrow$ to denote the reduction of $SC$ of SGDD to the GCond.
>
> | Methods | Ogbn-arxiv($r$=0.25%)  ($SC$/Acc.) | Reddit($r$=0.05%)  ($SC$/Acc.)) | YelpChi ($r$=0.10%) ($SC$/F1-macro) | Amazon($r$=0.2%) ($SC$/F1-macro) |
> | - | - | - | - | - |
> | GCond | 0.34/63.2 | 0.46/89.6 | 0.46/49.6 | 0.48/78.1 |
> | SGDD | 0.24/67.2 ↓29.4% |0.18/91.8 ↓51.1%|0.23/58.1 ↓50.0%|0.18/84.8 ↓62.5%|
>
> Conclusion(s):
>
> - **The** $SC$ **metric serves as a good indicator for measuring the relationship between LED shift and generalization performance.**

---

> > ### Author Response · Authors · 2023-08-17
> > **Further Discussions with Reviewer 5ivm**
> >
> > Dear reviewer 5ivm:
> >
> > Thanks again for pointing out our key benefits of preserving the structure information. According to your constructive comments, we made the necessary adjustments on introducing the optimal transport and conducted time-consuming experiments on $SC$. Additionally, we have revised our writing of relevant concepts to enhance the accessibility for readers.
> >
> > As the rebuttal period is about to close, may I know if our rebuttal addresses your concerns? Thank you for taking the time to review our work and provide your insightful comments.

---

> > > ### Author Response · Authors · 2023-08-20
> > > **Looking forward to your reply!**
> > >
> > > Dear reviewer 5ivm,
> > >
> > > As we are nearing the rebuttal deadline, may I know if you have any other concerns regarding our work? Thanks!
> > >
> > > Best,
> > >
> > > Author of Submission 1279

---

> > > ### Comment · Reviewer_5ivm · 2023-08-21
> > > **Thank you for the response**
> > >
> > > Thank the authors for the effort. My major concerns are basically solved, so I will raise the score to 5. However, the authors are recommended improving the writing/presentation carefully if accepted.

---

> > > > ### Author Response · Authors · 2023-08-21
> > > >
> > > > Dear Reviewer 5ivm, we sincerely appreciate your revised rating! We would like to meticulously refine our presentations and conduct a comprehensive proofreading of our work.
> > > >
> > > > Thanks again for your constructive feedback and support.

---

### Decision · Program_Chairs · 2023-09-21

**Decision:**

Accept (poster)

**Comment:**

This paper focuses on graph distillation and presents a structure-braodcasting strategy. Extensive experiments demonstrate its superiority. After the response, it receives mixed ratings, including three positive scores and two borderline reject. The main concerns lie in the inadequate comparison and insufficient experimental support for the cost. The authors provide detailed responses to these disadvantages. Overall, I think the improvement over recent methods is acceptable. I vote for its acceptance. Please incorporate these results in the camera-ready version.